# Selfee, self-supervised features extraction of animal behaviors

Yinjun Jia[1,2]*, Shuaishuai Li[3,4], Xuan Guo[1,2], Bo Lei[1,2], Junqiang Hu[1], Xiao-Hong Xu[3,4], Wei Zhang[1,2]*

[1]School of Life Sciences, IDG/McGovern Institute for Brain Research, Tsinghua University, Beijing, China; [2]Tsinghua-Peking Center for Life Sciences, Beijing, China; [3]Institute of Neuroscience, State Key Laboratory of Neuroscience, Chinese Academy of Sciences Center for Excellence in Brain Science and Intelligence Technology, Shanghai, China; [4]Shanghai Center for Brain Science and Brain-Inspired Intelligence Technology, Shanghai, China

**Abstract** Fast and accurately characterizing animal behaviors is crucial for neuroscience research. Deep learning models are efficiently used in laboratories for behavior analysis. However, it has not been achieved to use an end-to-end unsupervised neural network to extract comprehensive and discriminative features directly from social behavior video frames for annotation and analysis purposes. Here, we report a self-supervised feature extraction (Selfee) convolutional neural network with multiple downstream applications to process video frames of animal behavior in an end-to-end way. Visualization and classification of the extracted features (Meta-representations) validate that Selfee processes animal behaviors in a way similar to human perception. We demonstrate that Meta-representations can be efficiently used to detect anomalous behaviors that are indiscernible to human observation and hint in-depth analysis. Furthermore, time-series analyses of Meta-representations reveal the temporal dynamics of animal behaviors. In conclusion, we present a self-supervised learning approach to extract comprehensive and discriminative features directly from raw video recordings of animal behaviors and demonstrate its potential usage for various downstream applications.

*For correspondence:
jyj20@mails.tsinghua.edu.cn (YJ);
wei_zhang@mail.tsinghua.edu.cn (WZ)

**Competing interest:** The authors declare that no competing interests exist.

## Editor's evaluation

Jia et al., present a valuable machine learning framework, Selfee, based on deep neural networks for analyzing video recordings of animal behavior, which is efficient and runs in an unsupervised fashion, and requires very little pre-processing. Selfee should be of broad interest to researchers studying quantitative animal behavior.

## Introduction

Extracting representative features of animal behaviors has long been an important strategy for studying the relationship between genes, neural circuits, and behaviors. Traditionally, human observations and descriptions are the primary solutions for animal behavior analysis (*Hall, 1994*; *McGill, 1962*; *Rubenstein and Alcock, 2019*). Well-trained researchers would define a set of behavior patterns and compare their intensity or proportion between experimental and control groups. With the emergence and flourish of machine learning methodology, supervised learning has been assisting human annotations and achieved impressive results (*Jiang et al., 2019*; *Kabra et al., 2013*; *Segalin et al., 2020*). Nevertheless, supervised learning is limited by prior knowledge (especially used for feature

engineering) and manually assigned labels, thus could not identify behavioral features that are not annotated.

Other machine learning methods were then introduced to the field which was designed to extract representative features beyond human-defined labels. These methods can be generally divided into two major categories: one estimates animal postures with a group of pre-defined key points of the body parts, and the other directly transforms raw images. The former category marks representative key points of animal bodies, including limbs, joints, trunks, and/or other body parts of interest (*Graving et al., 2019*; *Günel et al., 2019*; *Mathis et al., 2018*). Those features are usually sufficient to represent animal behaviors. However, it has been demonstrated that the key points generated by pose estimation are less effective for direct behavior classification or two-dimensional visualization (*Luxem et al., 2020*; *Sun et al., 2021*). Sophisticated post-processing like recurrent neural networks (RNNs) (*Luxem et al., 2020*), non-locomotor movement decomposition (*Huang et al., 2021*), or feature engineering (*Sun et al., 2021*) can be applied to transform the key points into higher-level discriminative features. Additionally, the neglection of body parts could cause problems. For example, the position of the proboscis of a fly is commonly neglected in behavior studies using pose estimation software (*Calhoun et al., 2019*; *Sun et al., 2021*). Still, it is crucial for feeding (*Zhou et al., 2019*), licking behavior during courtship (*Mezzera et al., 2020*), and hardness detection for a substrate (*Zhang et al., 2020*). Finally, best to our knowledge, there is no demonstration of these pose-estimation methods applied to multiple animals of the same color with intensive interactions. Thus, the application of pose estimation to mating behaviors of two black mice, a broadly adopted behavior paradigm (*Bayless et al., 2019*; *Wei et al., 2018*; *Zhang et al., 2021*), could be limited because labeling body parts during mice mounting is challenging even for humans (see Discussion for more details). Therefore, using these feature extraction methods requires rigorously controlled experimental settings, additional feature engineering, and considerable prior knowledge of particular behaviors.

In contrast, the other category transforms pixel-level information without key point labeling, thus retaining more details and requiring less prior knowledge. Feature extraction of images could be achieved by wavelet transforms (*Wiltschko et al., 2015*) or Radon transforms *Berman et al., 2014*; *Ravbar et al., 2019* followed by principal component analysis (PCA), and these transforms can be applied to either 2D images or depth images. However, preprocessing such as segmentation and/or registration of the images is required to achieve spatial invariance in some implementations, a task that is particularly difficult for multi-agent videos (one method named ABRS solved this problem by using the spectra of Radon transforms, which is translation and rotation invariant; *Ravbar et al., 2019*). Beyond translation or rotation invariance, the relative position between animals is also important to social behaviors. Animals could be of varies of relative positions when perform the same type of behaviors. Extracted features should be invariant to these variations and capture the major characteristics. Additionally, because these methods usually use unlearnable transforms, although it makes them highly transferrable from dataset to dataset, they could not be adaptive to images with different characteristics and thus select the most discriminative and relevant features across the dataset automatically. Flourished deep learning methods, especially convolutional neural networks (CNNs) (*Lecun et al., 1998*), could adaptively extract features from diversified datasets. Also, they have been proven more potent than classic computer vision algorithms like wavelet transforms (*Romero et al., 2009*) and Radon transforms (*Aradhya et al., 2007*) on a famous grayscale dataset MNIST, even without supervising (*Ji et al., 2019*). Therefore, we attempt to adopt CNNs to achieve end-to-end feature extractions of animal behaviors that are comprehensive and discriminative.

The cutting-edge self-supervised deep learning methods aim to extract representative features for downstream missions by comparing different augmentations of the same image and/or different images (*Caron et al., 2020*; *Chen et al., 2020*; *Grill et al., 2020*; *He et al., 2020*; *Wu et al., 2018*). Compared with previous techniques, these methods have three major advantages. First, self-supervised or unsupervised methods could completely avoid human biases. Second, the augmentations used to create positive samples promise invariance of the neural networks to object sizes, spatial orientations, and ambient laminations so that registration or other preprocessing is not required. Finally, the networks are optimized to export similar results for positive samples and separate negative ones, such that the extracted features are inherently discriminative. Even without negative samples, the networks can utilize differential information within batches to obtain remarkable results on downstream missions like classification or image segmentation (*Chen and He, 2021*; *Grill et al., 2020*;

*Zbontar et al., 2021*). These advances in self-supervised learning provide a promising way to analyze animal behaviors.

In this work, we develop Selfee (**Sel**f-supervised **Fe**atures **E**xtraction) that adopts recently published self-supervised learning algorithms and CNNs to analyze animal behaviors. Selfee is trained on massive unlabeled behavior video frames (around five million frames from hundreds of videos) to avoid human bias in annotating animal behaviors, and it could capture a global character of animal behaviors even when detailed postures are hard to extract, similar to human perception. During the training process, Selfee learns to project images to a low-dimensional space without being affected by shooting conditions, image translation, and rotation, where cosine distance is proper to measure the similarities of original pictures. Selfee also provides potential for various downstream analyses. We demonstrate that the extracted features are suitable for t-SNE visualization, *k*-NN-based classification, *k*-NN-based anomaly detection, and dynamic time warping (DTW). We also show that further integrated modeling, like the autoregressive hidden Markov model (AR-HMM), is compatible with Selfee extracted Meta-representations. We apply Selfee to fruit flies, mice, and rats, three widely used model animals, and validate our results with manual annotations or pre-existed animal tracking methods. Discoveries of behavioral phenotypes in mutant flies by Selfee are consistent with either human observations or other animal tracking analysis, and can be validated by biological experiments. The performance of Selfee on these model species indicates its potential usage for behavioral studies of non-model animals as well as other tasks. We also provide an open-source Python project and pre-trained models of flies and mice to the community (see more in Code Availability).

## Results
### Workflow of Selfee and its downstream analyses

Selfee is trained to generate Meta-representations at the frame level, then analyzed at different time scales. First, grayscale videos are decomposed into single frames, and three tandem frames are stacked into a live-frame to generate a motion-colored RGB picture (*Figure 1A*). These live-frames preserve not only spatial information (e.g., postures of each individual or relative distances and angles between individuals) within each

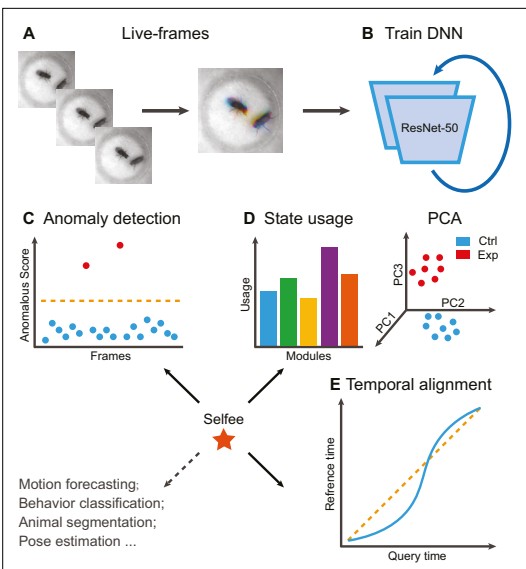

**Figure 1.** The framework of Selfee (**Sel**f-supervised **Fe**atures **E**xtraction) and its downstream applications. (**A**) One live-frame is composed of three tandem frames in R, G, and B channels, respectively. The live-frame could capture the dynamics of animal behaviors. (**B**) Live-frames are used to train Selfee, which adopts a backbone of ResNet-50. (C, D, and E) Representations produced by Selfee could be used for anomaly detection that could identify unusual animal postures in the query video compared with the reference videos. (**C**) AR-HMM (autoregressive hidden Markov model) that models the local temporal characteristics of behaviors and clusters frames into modules (states) and calculates stages usages of different genotypes (**D**) DTW (dynamic time warping) that aligns behavior videos to reveal differences of long-term dynamics (**E**) and other potential tasks including behavior classification, forecasting, or even image segmentation and pose estimation after appropriately modifying and fine-tuning of the neural networks.

The online version of this article includes the following video and figure supplement(s) for figure 1:

**Figure supplement 1.** Beddings and backgrounds that affect training and inference of Selfee (**Sel**f-supervised **Fe**atures **E**xtraction).

**Figure supplement 2.** t-SNE visualization of pose estimation derived features.

**Figure supplement 3.** Animal tracking with DLC, FlyTracker, and SLEAP.

**Figure 1—video 1.** Visualization of DLC tracking results on intensive interactions between mice during mating behavior.
https://elifesciences.org/articles/76218/figures#fig1video1

**Figure 1—video 2.** A tracking example of FlyTracker of Fly-vs-Fly dataset.

*Figure 1 continued on next page*

*Figure 1 continued*

https://elifesciences.org/articles/76218/figures#fig1video2

**Figure 1—video 3.** A tracking example of SLEAP on fly courtship behavior.

https://elifesciences.org/articles/76218/figures#fig1video3

channel but also temporal information across different channels. Live-frames are used to train Selfee to produce comprehensive and discriminative representations at the frame level (*Figure 1B*). These representations can be later used in numerous applications. For example, anomaly detection on mutant animals can discover new phenotypes compared with their genetic controls (*Figure 1C*). Also, the AR-HMM could be applied to model the micro-dynamics of behaviors, such as the duration of states or the probabilities of state transitions (*Wiltschko et al., 2015*). The AR-HMM splits videos into modules and yields behavioral state usages that visualize differences between genotypes (*Figure 1D*). In contrast, DTW could compare the long-term dynamics of animal behaviors and capture global differences at the video level (*Myers et al., 1980*) by aligning pairs of time series and calculating their similarities (*Figure 1E*). These three demonstrations cover different time scales from frame to video level, and other downstream analyses could also be incorporated into the workflow of Selfee.

Compared with previous machine learning frameworks for animal behavior analysis, Selfee has three major advantages. First, Selfee and the Meta-representations could be used for various tasks. The contrastive learning process of Selfee would allow output features to be appropriately compared by cosine similarity. Therefore, distance-based applications, including classification, clustering, and anomaly detection, would be easily realized. It was also reported that with some adjustment of backbones, self-supervised learning would facilitate tasks such as pose estimation (*Dahiya et al., 2021*) and object segmentation (*Caron et al., 2021*; *He et al., 2020*). Those findings indicate that Selfee could be generalized, modified, and fine-tuned for animal pose estimation or segmentation tasks. Second, Selfee is a fully unsupervised method developed to annotate animal behaviors. Although some other techniques also adopt semi-supervised or unsupervised learning, they usually require manually labeled pre-defined key points of the images (*Huang et al., 2021*; *Luxem et al., 2020*); some methods also require expert-defined programs for better performance (*Sun et al., 2021*). Key point selection and program incorporation require a significant amount of prior knowledge and are subject to human bias. In contrast, Selfee does not need any prior knowledge. Finally, Selfee is relatively hardware-inexpensive compared with other self-supervised learning methods (*Chen et al., 2020*; *Chen et al., 2020*; *Grill et al., 2020*). Training Selfee only takes 8 hr on a single RTX 3090 GPU (graphic card), and the inference speed could reach 800 frames per second. Selfee could accept top-view 2D grayscale video frames as inputs so that neither depth cameras (*Wiltschko et al., 2015*) nor fine-calibrated multi-view camera arrays (*Huang et al., 2021*) are required. Therefore, Selfee can be trained and used with routinely collected behavior videos on ordinary desktop workstations, warranting its accessibility to biology laboratories.

## Siamese CNNs capture discriminative representations of animal posture

Selfee contains a pair of Siamese CNNs trained to generate discriminative representations for live-frames. ResNet-50 (*He et al., 2016*) is chosen as the backbone whose classifier layer is replaced by a three-layer multi-layer perceptron (MLP). These MLPs are called projectors which yield final representations during the inference stage. There are two branches in Selfee. The main branch is equipped with an additional predictor, while the reference branch is a copy of the main branch (the SimSiam style; *Chen et al., 2020*). To the best of our knowledge, the SimSiam style is the most straightforward Siamese CNN frameworks with only one term of loss. Both branches contain group discriminators after projectors and perform dimension reduction on extracted features for online clustering (*Figure 2B*).

During the training stage, batches of live-frames are randomly transformed twice and fed into the main branch and reference branch, respectively. Augmentations applied to live-frames include crop, rotation, flip, and application of the Turbo lookup table (*Mikhailov, 2019*) followed by color jitters (*Figure 2A*, *Figure 2—figure supplement 1*). The reference branch yields a representation of received frames, while the main branch predicts the outcome of the reference branch. This is the first objective of the training process, which optimizes the cosine similarity between the outcome of the reference

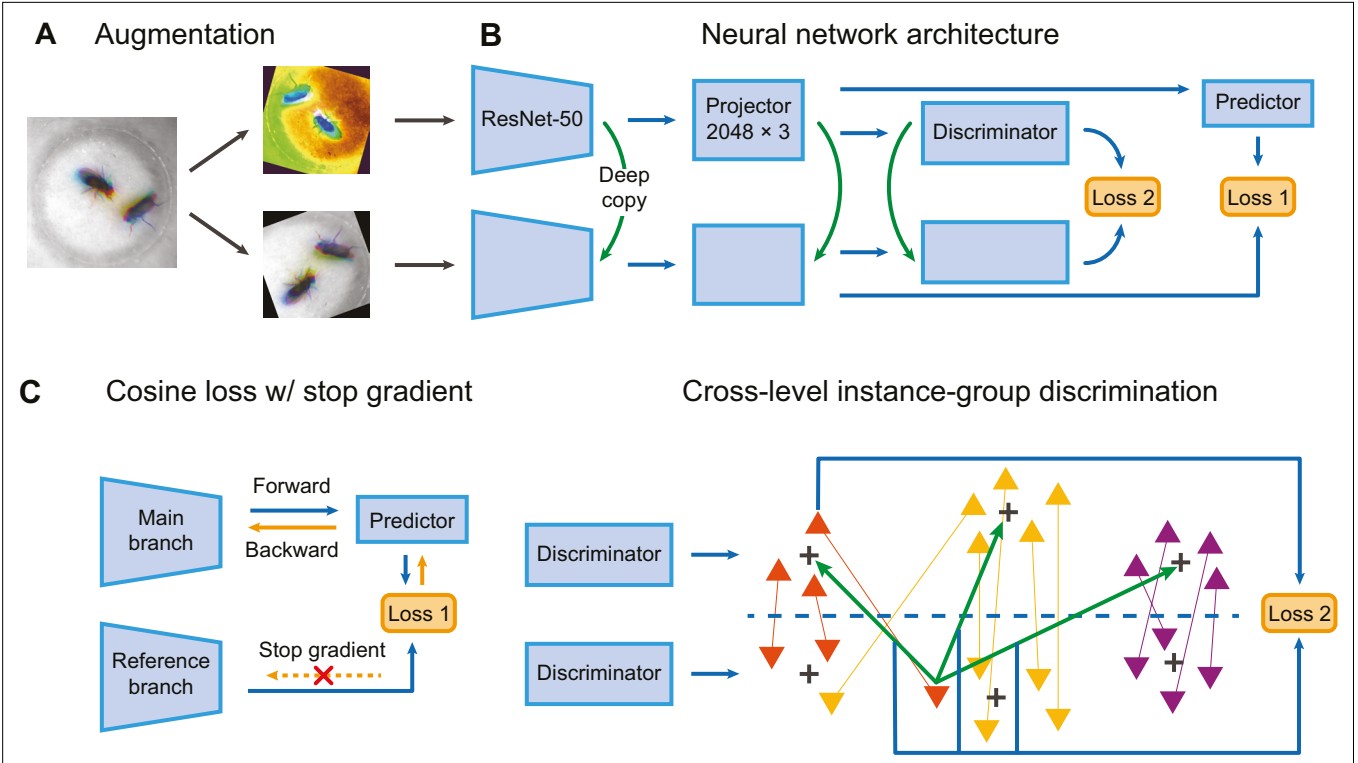

**Figure 2.** The network structure of Selfee (Self-supervised Features Extraction). (**A**) The architecture of Selfee networks. Each live-frame is randomly transformed twice before being fed into Selfee. Data augmentations include crop, rotation, flip, Turbo, and color jitter. (**B**) Selfee adopts a SimSiam-style network structure with additional group discriminators. Loss 1 is canonical negative cosine loss, and loss 2 is the newly proposed CLD (cross-level instance-group discrimination) loss. (**C**) A brief illustration of two loss terms used in Selfee. The first term of loss is negative cosine loss, and the outcome from the reference branch is detached from the computational graph to prevent mode collapse. The second term of loss is the CLD loss. All data points are colored based on the clustering result of the upper branch, and points representing the same instance are attached by lines. For one instance, the orange triangle, its representation from one branch is compared with cluster centroids of another branch and yields affinities (green arrows). Loss 2 is calculated as the cross-entropy between the affinity vector and the cluster label of its counterpart (blue arrows).

The online version of this article includes the following figure supplement(s) for figure 2:

**Figure supplement 1.** Different augmentations used for Selfee (Self-supervised Features Extraction) training.

branch and its prediction given by the main branch. To prevent mode collapse, the reference branch will not receive gradient information during optimization, which means the outcome of the reference branch is detached from the computational graph and treated as ground truth (*Figure 2C*). The second objective is to optimize the clustering results of representations from two discriminators. For one instance, its representation from one branch is compared with cluster centroids of another branch and yields affinities. The affinity vector is optimized to be consistent with the cluster label of its counterpart. In the original publication (*Wang et al., 2021*), this loss is termed cross-level instance-group discrimination (CLD) loss (*Figure 2C*). Because the main branch and reference branch are symmetric, similar loss calculations can also be done after swapping the identity of these two branches. The final loss is the average loss under the two conditions. In this way, Selfee is trained to be invariant to those transforms and focus on critical information to yield discriminative representations.

After the training stage, we evaluated the performance of Selfee with t-SNE visualization and *k*-NN classification. To investigate whether our model captured human-interpretable features, we manually labeled one clip of *Drosophila* courtship video and visualized those representations with t-SNE dimension reduction. On the t-SNE map, human-annotated courtship behaviors, including chasing, wing extension, copulation attempt, copulation, and non-interactive behaviors ('others'), were grouped in a non-random pattern (*Figure 3A*). Then, we would like to know if our neural network could capture fine-grained features beyond human-defined labels. Using cutting-edge animal tracking software SLEAP (*Pereira et al., 2022*), flies' wings, heads, tails, and thoraxes were tracked throughout the clip automatically with manual proofreading (*Figure 3—video 1*). Three straight-forward features were

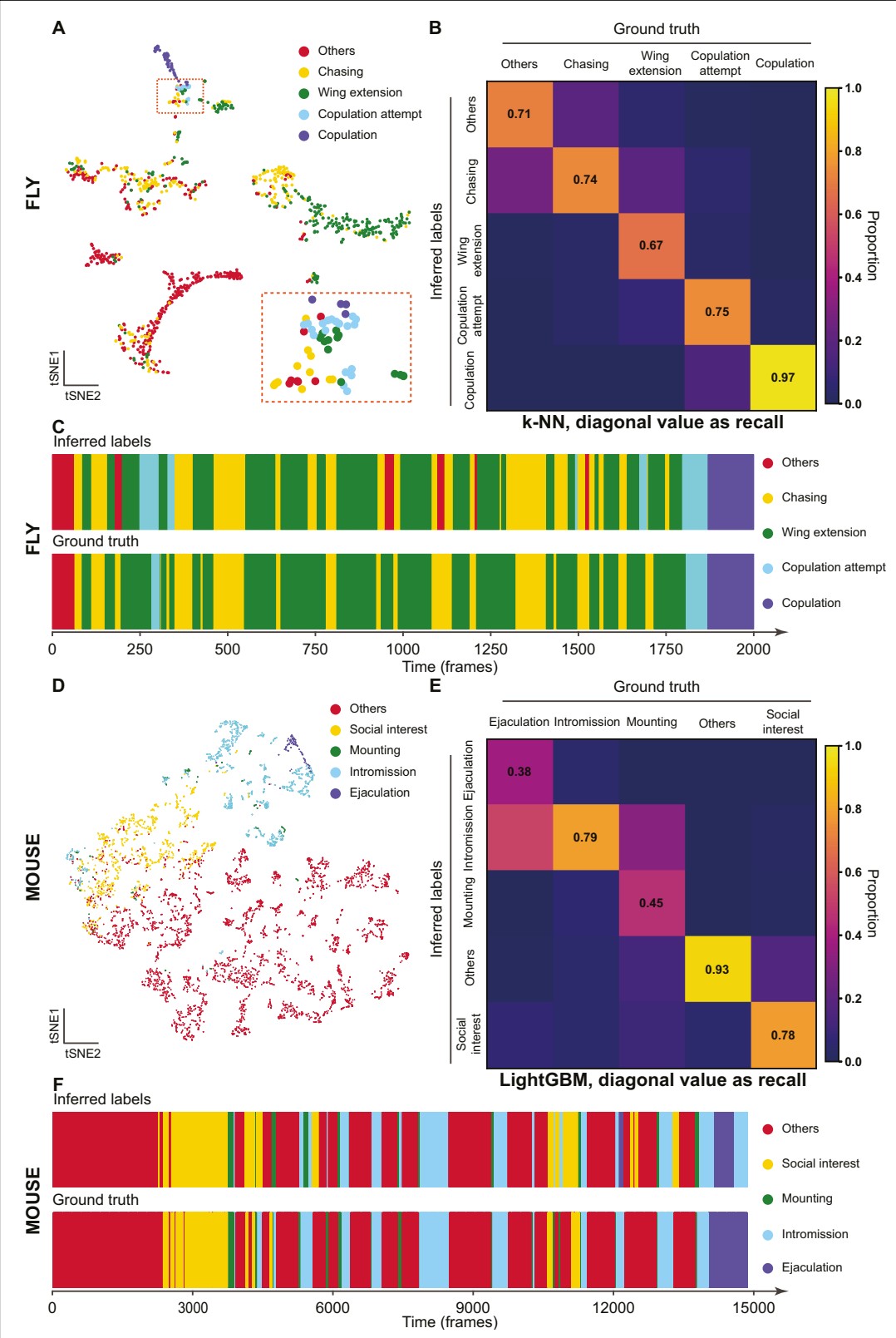

**Figure 3.** The validation of Selfee (<u>Sel</u>f-supervised <u>F</u>eatures <u>E</u>xtraction) with human annotations. (**A**) Visualization of fly courtship live-frames with t-SNE dimension reduction. Each dot was colored based on human annotations. Points representing chasing, wing extension, copulation attempt, copulation, and non-interactive behaviors ('others') were colored with yellow, green, blue, violet and red, respectively. (**B**) The confusion matrix of the *k*-NN classifier for fly courtship behavior, normalized by the numbers of each behavior in the ground truth. The average $F_1$ score of the sevenfold cross-validation was

*Figure 3 continued on next page*

*Figure 3 continued*

72.4%, and mAP was 75.8%. The recall of each class of behaviors was indicated on the diagonal of the confusion matrix. (**C**) A visualized comparison of labels produced by the *k*-NN classifier and human annotations of fly courtship behaviors. The *k*-NN classifier was constructed with data and labels of all seven videos used in the cross-validation, and the $F_1$ score was 76.1% and mAP was 76.1%. (**D**) Visualization of live-frames of mice mating behaviors with t-SNE dimension reduction. Each dot is colored based on human annotations. Points representing non-interactive behaviors ('others'), social interest, mounting, intromission, and ejaculation were colored with red, yellow, green, blue, and violet, respectively. (**E**) The confusion matrix of the LightGBM (Light Gradient Boosting Machine) classifier for mice mating behaviors, normalized by the numbers of each behavior in the ground truth. For the LightGBM classifier, the average $F_1$ score of the eightfold cross-validation was 67.4%, and mAP was 69.1%. The recall of each class of behaviors was indicated on the diagonal of the confusion matrix. (**F**) A visualized comparison of labels produced by the LightGBM classifier and human annotations of mice mating behaviors. An ensemble of eight trained LightGBM was used, and the $F_1$ sore was 68.1% and mAP was not available for this ensembled classifier due to the voting mechanism.

The online version of this article includes the following video and figure supplement(s) for figure 3:

**Figure supplement 1.** Selfee (<u>Sel</u>f-supervised <u>F</u>eatures <u>E</u>xtraction) captured fine-grained features related to animal postures and positions.

**Figure supplement 2.** Difficulties on fly courtship behavior classification.

**Figure supplement 3.** Classification of mice mating behaviors with Selfee (<u>Sel</u>f-supervised <u>F</u>eatures <u>E</u>xtraction) extracted features.

**Figure supplement 4.** *k*-NN classification of rat behaviors with Selfee (<u>Sel</u>f-supervised <u>F</u>eatures <u>E</u>xtraction) trained on mice datasets.

**Figure supplement 5.** Ablation test of Selfee (<u>Sel</u>f-supervised <u>F</u>eatures <u>E</u>xtraction) training process on fly datasets.

**Figure 3—video 1.** Pose estimation of fly courtship behaviors.

https://elifesciences.org/articles/76218/figures#fig3video1

visualized on the t-SNE map. Distances between male heads and female tails could indicate chasing intensity; wing angles of male flies were correlated to wing extension behavior, and distances of male flies away from the chamber center could reflect the trade-off between their thigmotaxis (*Besson and Martin, 2005*) and courtship motivation. We found that the features extracted by Selfee separated wing extension behaviors based on the wing angles (*Figure 3—figure supplement 1C*) and male head to female tail distance (*Figure 3—figure supplement 1D*). The result also showed that chasing behavior could be separated based on the positions of male flies in the chamber, which might indicate the thigmotaxis level. In conclusion, Selfee is capable of extracting comprehensive features that consist of human observations or animal tracking results, and in this way, Selfee uniforms natural-languages-based human descriptions and engineered features from tracking results in different units (e.g., rad for angle and mm for distance) and scales in a single discriminative Meta-representation.

Meta-representations can also be used for behavior classification. We manually labeled seven 10,000-frame videos (around 5 min each) as a pilot dataset. A weighed *k*-NN classifier was then constructed as previously reported (*Wu et al., 2018*). Sevenfold cross-validation was performed on the dataset with the *k*-NN classifier, which achieved a mean $F_1$ score of 72.4% and achieved a similar classification result as human annotations (*Figure 3B and C*). The classifier had the worst recall score on wing extension behaviors (67% recall, *Figure 3B*), likely because of the ambiguous intermediate states between chasing and wing extension (*Figure 3—figure supplement 2A*). The precisions also showed that this *k*-NN classifier tended to have strict criteria for wing extension and copulation and relatively loose criteria for chasing and copulation attempts (*Figure 3—figure supplement 2B*). It was reported that independent human experts could only reach agreements on around 70% of wing extension frames (*Leng et al., 2020*), comparable to the performance of our *k*-NN classifier.

Next, we compared Selfee extracted features with animal-tracking derived features. In a previous study *Leng et al., 2020*, labeled wing extension intensities of male flies for three clips of male-female interaction videos: each frame was scored from 0 (no wing extension) to 3 (strong

**Table 1.** A comparison between Selfee (<u>Sel</u>f-supervised <u>F</u>eatures <u>E</u>xtraction) extracted features and animal-tracking derived features.

| Evaluations setups | Pearson's R | $F_1$ score | AP |
|---|---|---|---|
| Selfee | **0.774**[*] | **0.629**[*] | 0.354 |
| FlyTracker ->FlyTracker | 0.756 | 0.571 | 0.330 |
| FlyTracker ->JAABA | 0.755 | 0.613 | 0.346 |
| FlyTracker (w/o legs) ->distance | 0.771 | 0.613 | **0.374***  |
| FlyTracker (w/ legs) ->distance | 0.629 | 0.400 | 0.256 |

*Best results of different feature extractors under each evaluation metric are indicated in bold values.

wing extension) by two experienced researchers. Here, we used their summarized score as ground-truth labels. FlyTracker (*Fleet et al., 2014*) was used to track each fly's body, wings and legs, and the identity swap between male and female flies was manually corrected in the previous work. We used four types of post-processing for the tracking results. Features from FlyTracker and JAABA (*Kabra et al., 2013*) were from the work by *Leng et al., 2020*. We also constructed pure distance-based features recording distances between all key points (heads, tails, thoraxes, wings, and with or without legs). For a fair comparison, we evaluated the performance of weight $k$-NN classifiers in sixfold cross-validations for all types of features, and none of the additional temporal information aggregation was used (e.g., sliding window voting used in Selfee, bout features used in FlyTracker, or window features used in JAABA). Three evaluation metrics were applied, including a robust version of Pearson's correlation coefficient (*Lai et al., 2019*), $F_1$ score, and average precision. We found that Selfee extracted features achieved comparable results with FlyTracker features or JAABA features, and it achieved the best performance evaluated by Pearson's correlation and $F_1$ score (*Table 1*). We also found that additional irrelevant key points marking fly legs would strongly interfere the performance of pure distance-based features, indicating that key point choice and feature engineering were crucial for downstream classification. The comparison also yielded an interesting result that the pure distance-based feature could even outperform human-engineered features like FlyTracker features. Distances between key points indeed preserved detailed information on animal behaviors, but it remained unclear if they could capture the universals of behavioral stereotypes. For further investigation, we visualized distance-based features and human-engineered features on the same clip as in *Figure 3A* with t-SNE dimension reduction. Results showed that distance-based features were overfocused on subtle differences between frames and neglected differences between major types of behaviors. In contrast, human-engineered features formed distinct clusters corresponding to human annotations on the t-SNE map (*Figure 1—figure supplement 2*), so did Selfee features (*Figure 3A*). Therefore, although pure distance-based features could outperform human-engineered features and Selfee features using highly non-linear $k$-NN classifier, they were less abstractive. Overall, Selfee extracted features are as discriminative as classic animal-tracking derived features, but could be used more easily without careful key points definition or dedicated feature engineering.

We then asked whether Selfee can be generalized to analyze behaviors of other species. We fine-tuned fly video pre-trained Selfee with mice mating behavior data. The mating behavior of mice can be defined mainly into five categories (*McGill, 1962*), including social interest, mounting, intromission, ejaculation, and others (see Materials and methods for detailed definitions). With t-SNE visualization, we found that five types of behaviors could be separated by Selfee, although mounting behaviors were rare and not concentrated (*Figure 3D*). We then used eight human-annotated videos to test the $k$-NN classification performance of Selfee-generated features. We achieved an $F_1$ score of 59.0% (Table 3-Replication 1, *Figure 3—figure supplement 3*). Mounting, intromission, and ejaculation share similar static characteristics but are different in temporal dynamics. Therefore, we asked if more temporal information would assist the classification. Using the LightGBM (Light Gradient Boosting Machine) classifier (*Ke et al., 2017*), we achieved a much higher classification performance by incorporating slide moving average and standard deviation of 81-frame time windows, the main

**Table 2.** An ablation test of Selfee (<u>Sel</u>f-supervised <u>Fe</u>atures <u>E</u>xtraction) training process on fly datasets.

| Model | Pre-trained ResNet-50 with random projectors | | Selfee | | Selfee without CLD loss | | Selfee without Turbo transformation | |
|---|---|---|---|---|---|---|---|---|
| Evaluation | Mean $F_1$ score | Mean AP | Mean $F_1$ score | Mean AP | Mean $F_1$ score | Mean AP | Mean $F_1$ score | Mean AP |
| Replication 1 | 0.586 | 0.580 | 0.724 | 0.758 | 0.227 | 0.227 | 0.604 | 0.550 |
| Replication 2 | 0.597 | 0.570 | 0.676 | 0.683 | 0.163 | 0.200 | 0.574 | 0.551 |
| Replication 3 | 0.596 | 0.586 | 0.714 | 0.754 | 0.172 | 0.214 | 0.517 | 0.497 |
| Best | 0.597 | 0.586 | **0.724**[*] | **0.758**[*] | 0.227 | 0.227 | 0.604 | 0.551 |

[*]Best results of different training setups under each evaluation metric are indicated in bold values.

**Table 3.** An ablation test of Selfee training process on mice datasets.

| Model | Single frame + KNN | | Live-frame + KNN | | Single frame + LGBM | | Live-frame + LGBM | |
|---|---|---|---|---|---|---|---|---|
| Evaluation | Mean $F_1$ score | Mean AP | Mean $F_1$ score | Mean AP | Mean $F_1$ score | Mean AP | Mean $F_1$ score | Mean AP |
| Replication 1 | 0.554 | 0.498 | 0.590 | 0.530 | 0.645 | 0.671 | 0.674 | 0.691 |
| Replication 2 | 0.574 | 0.508 | 0.599 | 0.549 | 0.653 | 0.663 | 0.663 | 0.699 |
| Replication 3 | 0.566 | 0.514 | 0.601 | 0.539 | 0.652 | 0.692 | 0.663 | 0.700 |
| Mean | 0.565 | 0.507 | 0.597 | 0.539 | 0.650 | 0.675 | **0.667**[*] | **0.697**[*] |
| Best | 0.574 | 0.514 | 0.601 | 0.549 | 0.653 | 0.692 | **0.674**[*] | **0.700**[*] |

*Best results of different training setups under each evaluation metric are indicated in bold values.

frequencies, and their energy within 81-frame time windows. The average $F_1$ score of eightfold cross-validation could reach 67.4%, and the classification results of the ensembled classifier (see Materials and methods) were closed to human observations (*Figure 3E and F*). Nevertheless, it was still difficult to distinguish between mounting, intromission, and ejaculation because mounting and ejaculation are much rarer than social body contact or intromission.

Selfee is more robust than the vanilla SimSiam networks when applied to the behavioral data. Behavioral data often suffer from severe imbalance. For example, copulation attempts are around sixfold rarer than wing extension during fly courtship (*Figure 3—figure supplement 5A*). Therefore, we added group discriminators to vanilla SimSiam networks, which were reported to fight against the long-tail effect proficiently (*Wang et al., 2021*). As noted in the original publication of CLD loss, $k$-means clustering lifted weights of rare behaviors in batches from the reciprocal of the batch size to the reciprocal of the cluster number, for which the imbalance between majority and minority classes can be ameliorated (*Wang et al., 2021*). Aside from overcoming the long-tail effect, we also found group discriminators helpful for preventing mode collapse during ablation studies (*Figure 3—figure supplement 5B-D* and *Table 2*). We hypothesized that the convergence could be easily reached on images of similar objects (two flies), by which CNNs may not be well trained to extract good representations. Without CLD loss, it could be easier to output similar representations than to distinguish images apart, because only the attraction between positive samples was applied. Using CLD loss, Selfee could explicitly utilize negative samples and encourage both repulsion and attraction. Therefore, the mode collapse could be largely avoided. Aside from CLD loss, Turbo transformation was another customed modification. Applying the Turbo lookup table on grayscale frames brought more complexity and made color distortions more powerful on grayscale images, one of the most critical type of augmentations reported before (*Chen et al., 2020*). Because hue and saturation of a grayscale image are meaningless, color jitter can be strongly enhanced after Turbo transforms. Selfee would capture more useful features with this Turbo augmentation (*Figure 3—figure supplement 5E, F* and *Table 2*). Live-frames used by Selfee were also helpful to capture temporal information of animal behaviors, especially for highly dynamic ones such as mice mating behaviors. We found that Selfee features extracted from single frames were less discriminative than those extracted from live-frames (*Table 3*). In summary, our modifications to original SimSiam designs, including live-frames extracting temporal dynamics, Turbo transformation customed for grayscale images, and CLD loss coping with the long-tailed nature of behavioral data, provide a significant performance boost in the case of animal behavior analysis.

## Anomaly detection at the frame level identifies rare behaviors at the sub-second time scale

The representations produced by Selfee could be directly used for anomaly detection without further post-processing. During the training step, Selfee learns to compare Meta-representations of frames with cosine distance which is also used for anomaly detection. When given two groups of videos, namely the query group and the reference group, the anomaly score of each live-frame in the query group is calculated in two steps (*Figure 4A*). First, distances between the query live-frame and all reference live-frames are measured, and the $k$-nearest distance is referred to as its inter-group score (IES). Without further specification, $k$ equals 1 in all anomaly detections in this work, which is a trivial

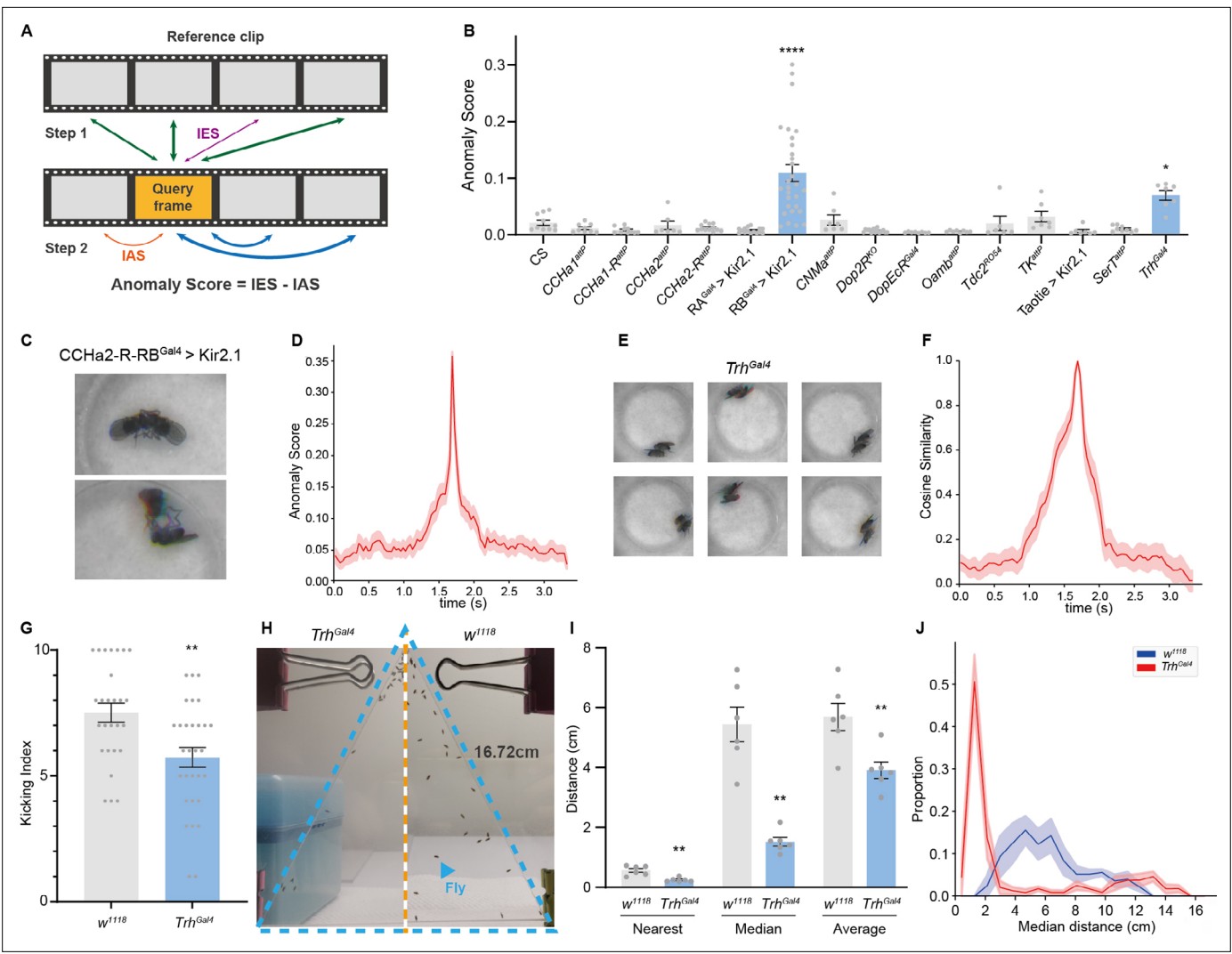

**Figure 4.** Anomalous posture detection using Selfee (Self-supervised Features Extraction)-produced features. (**A**) The calculation process of anomaly scores. Each query frame is compared with every reference frame, and the nearest distance was named IES (the thickness of lines indicates distances). Each query frame is also compared with every query frame, and the nearest distance is called IAS. The final anomaly score of each frame equals IES minus IAS. (**B**) Anomaly detection results of 15 fly lines with mutations in neurotransmitter genes or with specific neurons silenced ( n = 10,9,10,7,12,15,29,7,16,8,8,6,7,7,9,7, respectively). RA is short for CCHa2-R-RA, and RB is short for CCHa2-R-RB. CCHa2-R-RB^{Gal4}>Kir2.1, q<0.0001; Trh^{Gal4}, q=0.0432; one-way ANOVA with Benjamini and Hochberg correction. (**C**) Examples of mixed tussles and copulation attempts identified in CCHa2-R-RB^{Gal4}>Kir2.1 flies. (**D**) The temporal dynamic of anomaly scores during the mixed behavior, centralized at 1.67 s. SEM is indicated with the light color region. (**E**) Examples of close body contact behaviors identified in Trh^{Gal4} flies. (**F**) The cosine similarity between the center frame of the close body contact behaviors (1.67 s) and their local frames. SEM is indicated with the light color region. (**G**) The kicking index of Trh^{Gal4} flies (n=30) was significantly lower than w^{1118} flies (n=27), p=0.0034, Mann-Whitney test. (**H**) Examples of social aggregation behaviors of Trh^{Gal4} flies and w^{1118} flies. Forty male flies were transferred into a vertically placed triangle chamber (blue dashed lines), and the photo was taken after 20 min. A fly was indicated by a blue arrow. The lateral sides of the chamber were 16.72 cm. (**I**) Social distances of Trh^{Gal4} flies (n=6) and w^{1118} flies (n=6). Trh^{Gal4} flies had much closer social distances with each other compared with w^{1118} flies; nearest, p=0.0043; median, p=0.002; average, p=0.0087; all Mann-Whitney test. (**J**) Distributions of the median social distance of Trh^{Gal4} flies and w^{1118} flies. Distributions were calculated within each replication. Average distributions were indicated with solid lines, and SEMs were indicated with light color regions.

The online version of this article includes the following video and figure supplement(s) for figure 4:

**Figure supplement 1.** Using intra-group score (IAS) to eliminate false-positive results in anomaly detections.

**Figure 4—video 1.** The anomaly detection on male-male interactions of RB-Gal4 >Kir2.1 flies.

https://elifesciences.org/articles/76218/figures#fig4video1

**Figure 4—video 2.** The anomaly detection on male-male interactions of Trh^{Gal4} flies.

https://elifesciences.org/articles/76218/figures#fig4video2

and intuitive case of the *k*-NN algorithm to avoid parameter search of *k*. Some false positives occurred when only the IES was used as the anomaly score (*Figure 4—figure supplement 1A*). The reason could be that two flies in a chamber could be in mathematically infinite relative positions and form a vast event space. However, each group usually only contains several videos, and each video is only recorded for several minutes. For some rare postures, even though the probability of observing them is similar in both the query and reference group, they might only occur in the query group but not in the reference group. Therefore, an intra-group score (IAS) is introduced in the second step to eliminate these false-positive effects. We assume that those rare events should not be sampled frequently in the query groups either. Thus, the IAS is defined as the *k*-nearest distance of the query frame against all other frames within its group, except those within the time window of ±50 frames, because representations for frames beyond 50 frames were less similar to the current frame (*Figure 4—figure supplement 1B*). The final anomaly score is defined as the IES minus the IAS.

To test whether our methods could detect anomalous behavior in real-world data, we performed anomaly detection to 15 previously recorded neurotransmitter-related mutant alleles or neuron-silenced lines (with UAS-Kir2.1; *Paradis et al., 2001*; *Figure 4B*). Their male-male interaction videos were inferred by Selfee trained on male-female courtship videos. Since we aimed to find interactions distinct from male-male courtship behaviors, a baseline of ppk23 >Kir2.1 flies was established because this line exhibits strong male-male courtship behaviors (*Thistle et al., 2012*). We compared the top-100 anomaly scores from sets of videos from experimental groups and wild-type control flies. The results revealed that one line, CCHa2-R-RB>Kir2.1, showed a significantly high anomaly score. By manually going through all anomalous live-frames, we further identified its phenotype as a brief tussle behavior mixed with copulation attempts (*Figure 4C*, *Figure 4—video 1*, 0.2× play speed). This behavior was ultra-fast and lasted for less than a quarter second (*Figure 4D*), making it difficult to be detected by human observers. Up to this point, we have demonstrated that the frame-level anomaly detection could capture sub-second behavior episodes that human observers tend to neglect.

Selfee also revealed that *Trh* (*Tryptophan hydroxylase*) knock-out flies had close body contact compared to wild-type. *Trh* is the crucial enzyme for serotonin biosynthesis (*Coleman and Neck-ameyer, 2005*), and its mutant flies showed a statistically significantly higher anomaly score (*Figure 4B*) than the wild-type control. Selfee identified 60 frames of abnormal behaviors within 42,000 input frames, occupying less than 0.15% of the total recording time. By manually going through all these frames, we concluded most of them as short-range body interactions (*Figure 4E* and *Figure 4—video 2*, 0.2× play speed). These social interactions could last for around 1 s on average (*Figure 4F*). Even though serotonin signals were well studied for controlling aggression behavior in flies (*Alekseyenko et al., 2014*), to the best of our knowledge, the close body contact of flies and serotonergic neurons' role in this behavior has not been reported yet. Considering this behavior is not as fast as the ones of CCHa2-R-RB>Kir2.1 flies for humans, a possible reason is that this behavior is too scarce to be noticed by human experts.

To further ask whether these close body contacts have biological significance, we performed corresponded behavior assays on mutant flies. Based on the fact that the *Trh* mutant male flies have a higher tolerance to body touch, we hypothesized that they would have a decreased defensive behavior. As previously reported, fruit flies show robust defensive behavior to mechanical stimuli on their wings (*Li et al., 2016*; *Liu et al., 2020*). Decapitated flies would kick with their hind legs when a thin probe stimulates their wings. This stimulation mimics the invasion of parasitic mites and could be used to test its defensive behavior. Our results showed that *Trh* knock-out flies had a significantly lower kicking rate than control flies (*Figure 4G*), indicating a reduction of self-defensive intensity. Next, we performed social behavior assay (*McNeil et al., 2015*; *Simon et al., 2012*) on the mutant flies because the close body contact can also be explained by reduced social distance. We measured the nearest distance, median distance, and average distance of each male fly in a 40-individual group placed in a vertical triangular chamber (*Figure 4H*). By comparing median values of these distances of each replication, *Trh* knock-out flies kept significantly shorter distances from others than the control group (*Figure 4H, I*). The probability density function of their median distances also showed that knock-out flies had a closer social distance than control flies (*Figure 4J*). Therefore, we concluded that *Trh* knock-out flies had reduced self-defensive behavior and social distance, which validated the anomaly detected by Selfee. Taken together, Selfee is capable of discovering novel features of animal behaviors with biological relevance when a proper baseline is defined.

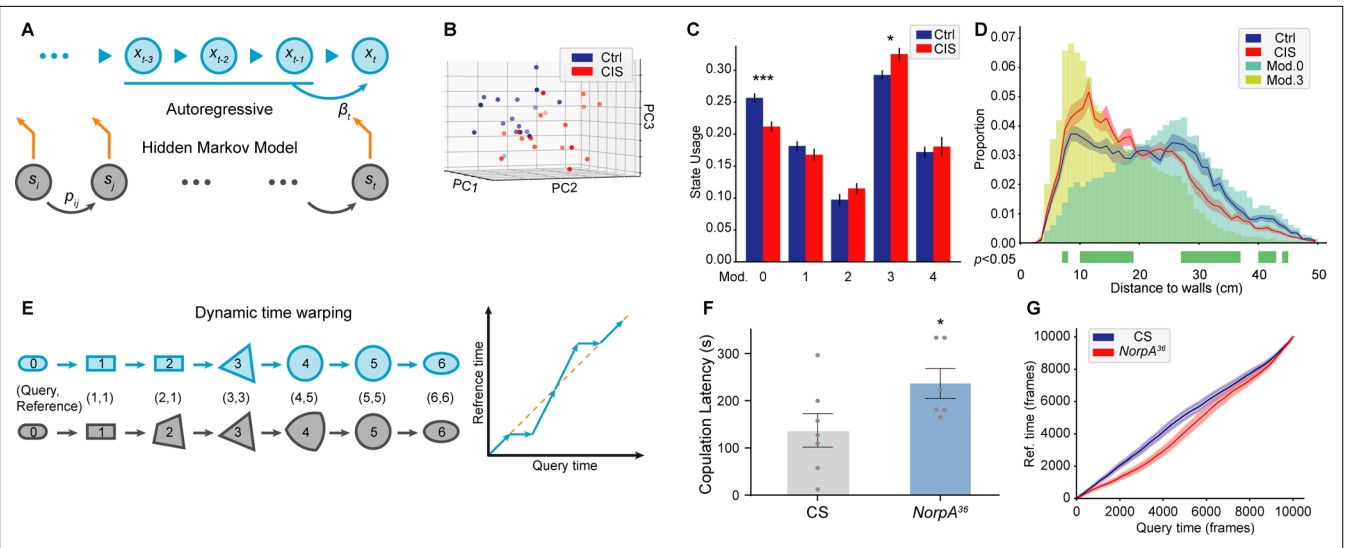

**Figure 5.** Time-series analyses using Selfee (Self-supervised Features Extraction)-produced features. (**A**) A brief illustration of the autoregressive hidden Markov model (AR-HMM). The local autoregressive property is determined by $\beta_t$, the autoregressive matrix, which is yield based on the current hidden state of the HMM. The transition between each hidden state is described by the transition matrix ($p_{ij}$). (**B**) Principal component analysis (PCA) visualization of state usages of mice in control groups (n=17, blue points) and chronic immobilization stress (CIS) groups (n=17, red points). (**C**) State usages of 10 modules. Module No.0 and No.3 showed significantly different usages in wild-type and mutant flies; p=0.00065, q=0.003 and p=0.015, q=0.038, respectively, Mann-Whitney test with Benjamini and Hochberg correction. (**D**) The differences spotted by the AR-HMM could be explained by the mice's position. Mice distances to the two nearest walls were calculated in each frame. Distance distributions (the bin width was 1 cm) throughout open-field test (OFT) experiments were plotted in solid lines, and SEMs were indicated with light color regions. Green blocks indicated bins with statistic differences between the CIS group and control groups. Frames assigned to modules No.0 and No.3 were isolated, and their distance distributions were plotted in blue and yellow bars, respectively. Frames of module No.0 were enriched in bins of larger values, while frames of module No.3 were enriched in bins of smaller values. (**E**) A brief illustration of the dynamic time warping (DTW) model. The transformation from a rounded rectangle to an ellipse could contain six steps (gray reference shapes). The query transformation lags at step 2 but surpasses at step 4. The dynamic is visualized on the right panel. (**F**) $NorpA^{36}$ flies (n=6) showed a significantly longer copulation latency than wild-type flies (n=7), p=0.0495, Mann-Whitney test. (**G**) $NorpA^{36}$ flies had delayed courtship dynamics than wild-type flies with DTW visualization. Dynamic of wild-type flies and $NorpA$ mutant flies were indicated by blue and red lines, respectively, and SEMs were indicated with light color regions. The red line was laid below the blue line, showing a delayed dynamic of $NorpA$ mutant flies.

The online version of this article includes the following video for figure 5:

**Figure 5—video 1.** A video example of Module No.0.

https://elifesciences.org/articles/76218/figures#fig5video1

**Figure 5—video 2.** A video example of Module No.3.

https://elifesciences.org/articles/76218/figures#fig5video2

## Modeling motion structure of animal behaviors

Animal behaviors have long-term structures beyond single-frame postures. The duration and proportions of each bout and transition probabilities of different behaviors have been proven to have biological significance (*Mueller et al., 2019*; *Wiltschko et al., 2015*). To better understand those long-term characteristics, we introduce AR-HMM and DTW analyses to model the temporal structure of animal behaviors. AR-HMM is a powerful method for analyzing stereotyped behavioral data (*Rudolph et al., 2020*; *Wiltschko et al., 2015*; *Wiltschko et al., 2020*). It discovers modules of behaviors and describes the modules with autoregressive matrixes. In other words, each embedding of the frame is predicted by a linear combination of embeddings of several previous frames, and frames within each module share the same coefficients of the linear combination, which is called autoregressive matrixes. In an AR-HMM, these fitted autoregressive patterns are utilized as hidden states of the HMM. The transition between each hidden state (autoregressive pattern) is determined by the transition matrix of the HMM. By the definition of Markov property, the transition from the current state to the next state is only determined by the current state and transition probabilities (*Figure 5A*). In this way, AR-HMM could capture local structures (autoregressive pattern) of animal behaviors as well as syntaxes (transition probabilities).

We asked if we could detect the dynamic changes in mice behaviors after chronic immobilization stress (CIS) during the open-field test (OFT). The CIS model is well established to study the depression-like behavior of experimental animals, and the anxiety level of mice can be evaluated with OFT. Mice prefer to walk near the wall, and the time spent in the center of the arena is considered to be related with anxiety behavior (*Crusio et al., 2013*; *Prut and Belzung, 2003*). We tested the OFT performance of mice with or without the CIS treatment. After preprocessing, videos were processed with Selfee trained with mice mating behavior. An AR-HMM with five modules (No.0 to No.4) was fitted to analyze behaviors during OFT. PCA of state usages revealed an apparent difference between mice with and without CIS experience (*Figure 5B*). Usages of two modules (No.0 and No.3) showed statistically significant differences between the two groups (*Figure 5C*). By watching sampled fragments of these two behaviors, we found module No.0 might be exploratory-like behavior, while module No.3 contained mice walking alongside walls (*Figure 5—videos 1 and 2*). To further confirm these observations, videos were analyzed by an animal tracking program, whose results were proofread manually. Mice in the CIS group spent more time near walls while mice in the control group spent more time in the central area (*Figure 5C*, blue and red lines), and similar observations had been reported before (*Ramirez et al., 2015*). Then, we analyzed whether these two modules were related to mice's position in the arena. All frames belonging to each module were extracted, and distances to the two nearest walls were calculated, the same as what was performed on all videos. The result indicated that mice performing behavior No.0 were relatively distant from walls, while in module No. 3, mice were closer to the border (*Figure 5C*, cyan and yellow bars). These results showed that Selfee with AR-HMM successfully distinguished mice in the control group from the CIS group, and the differences spotted by Selfee were consistent with previous observations (*Ramirez et al., 2015*). It is also established that Selfee with AR-HMM could discover the differences in proportions of behaviors, similar to what could be achieved with classic manual analysis or animal tracking software.

The AR-HMM modeling does not necessarily capture the difference in long-term dynamics intuitively, such as the latency of certain behaviors. To solve this problem, we introduce DTW analysis. DTW is a well-known algorithm for aligning time series, which returns the best-matched path and the matching similarity (*Figure 5E*). The alignment can be simplified as follows. When given the same start state and end state, it optimally maps all indices from the query series to the reference series monotonically. Pairs of mapped indices form a path to visualize the dynamic difference. The points above the diagonal line indicate that the current time point in the query group is matched to a future time point in the reference group so that the query group has faster dynamics and vice versa. Our experiments use cosine similarities of Selfee extracted representations to calculate warping paths.

Previously, DTW was widely applied to numerical measures of animal behaviors, including trajectory (*Cleasby et al., 2019*), audios (*Kohlsdorf et al., 2016*), and acceleration (*Aurasopon, 2016*). For the first time, we applied DTW to image data, with the aid of Selfee, to study the prolonged dynamic of animal behaviors. We applied DTW to analyze representations of *NorpA* mutant flies. Visual cues are essential for male flies to locate female flies during courtship (*Ribeiro et al., 2018*), and mutant flies of *NorpA*, which have defective visual transduction (*Bloomquist et al., 1988*), have a prolonged courtship latency in our experiments (*Figure 5F*), similar to previously findings (*Markow and Manning, 1980*). When wild-type flies were used as the reference for the DTW, the group of *NorpA* mutant flies yielded a curve lower than the diagonal line, indicating a delay in their courtship behaviors (*Figure 5G*). In this way, our experiments confirm that Selfee and DTW could capture differences in long-term dynamics such as behavior latency. In conclusion, DTW and AR-HMM could capture temporal differences between control and experimental groups beyond single-frame postures, making Selfee a competent unsupervised method for traditional analyses like courtship index or copulation latency.

## A brief demonstration of the whole Selfee pipeline

Selfee is a powerful end-to-end unsupervised behavior analysis tool. We went through the whole pipeline using the mice OFT discussed in previous sections as a demonstration. The first step to use Selfee is setting up a development environment containing packages in Key resources table. When recording conditions are consistent for different sets of videos, preprocessing can be simple frame extraction (*Figure 6A*, the dashed line). As mice open-field videos were recorded with some variations

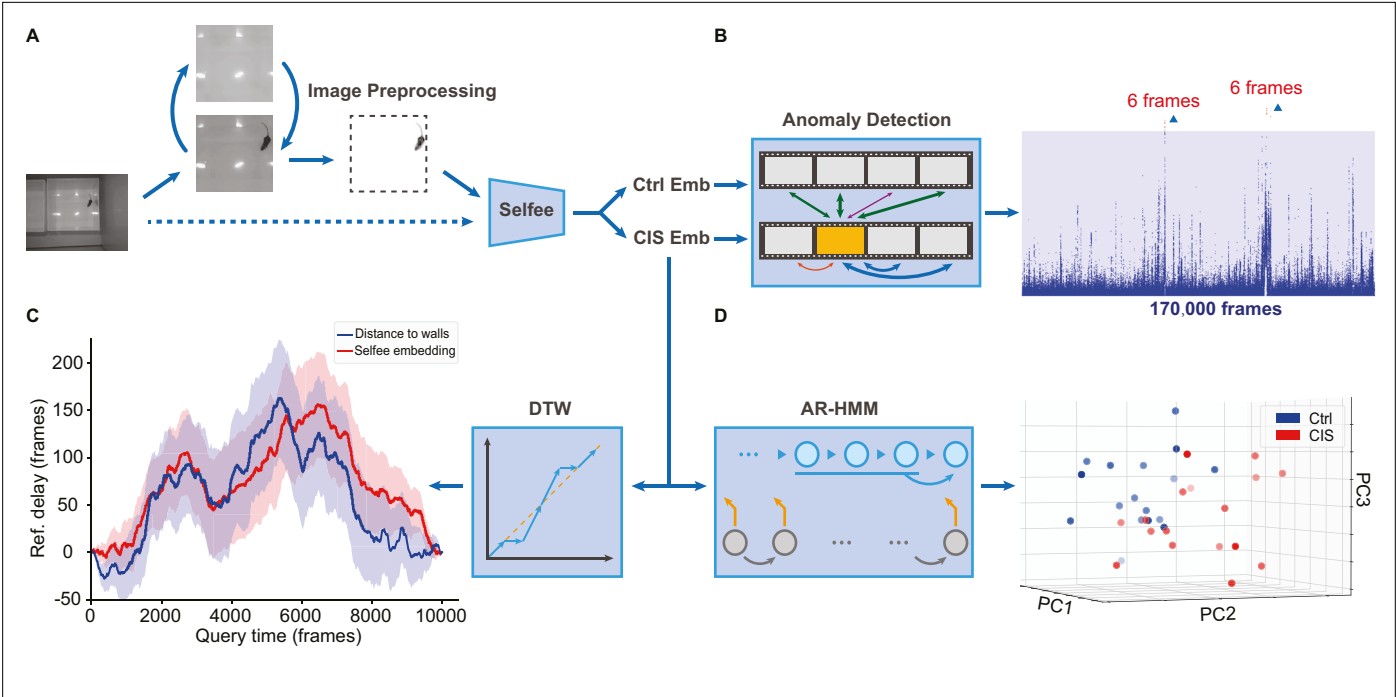

**Figure 6.** Application of the Selfee (Self-supervised Features Extraction) pipeline to mice open-field test (OFT) videos. (**A**) Image preprocessing for Selfee. The area of the behavior chamber was cropped, and the background was extracted. Illumination normalization was performed after background subtraction. This preprocessing could be skipped if the background was consistent in each video, as our pipeline for fly videos (dashed lines). (**B**) Anomaly detection of mice OFT videos after chronic immobilization stress (CIS) experiences. Only 12 frames (red points, indicated by arrows) were detected based on a threshold constructed with control mice (the blue region), and anomaly scores were slightly higher than the threshold. (**C**) Dynamic time warping (DTW) analysis of mice OFT videos after CIS experiences. The dynamic difference between control groups and CIS groups was visualized, and positive values indicated a delay of the reference (control groups). Results from Selfee features and animal positions were similar (red and blue lines, respectively). (**D**) Autoregressive hidden Markov model (AR-HMM) analysis of mice OFT videos after CIS experiences. Principal component analysis (PCA) visualization of state usages of mice in control groups (n=17, blue points) and CIS groups (n=17, red points). Same as *Figure 5B*.

The online version of this article includes the following figure supplement(s) for figure 6:

**Figure supplement 1.** Anomaly detection of chronic immobilization stress (CIS) mice.

**Figure supplement 2.** Dynamic time warping (DTW) analysis of chronic immobilization stress (CIS) and control mice.

in our case, arenas in the video were first cropped; backgrounds were removed, and the luminance was normalized (*Figure 6A*, solid lines).

After preprocessing, image embeddings were extracted with pre-trained Selfee. Features were then grouped according to experimental designs, and features for mice in the control group and CIS group were sent to the following modules. First, features of the control group were randomly assigned as references or negative controls. The maximum anomaly score of negative controls was set as the threshold. In the CIS group, 12 frames of anomaly behaviors were sorted out (*Figure 6A*, *Figure 6—figure supplement 1A*). However, these anomaly frames with their score slightly higher than the threshold only contributed to less than 0.01% of all frames, and occurred only in 2 out of 17 videos. Therefore, these frames were classified as false-positive after being examined manually (*Figure 6—figure supplement 1B*). Second, the features of the control group and CIS group were analyzed with AR-HMM. As previously showed, AR-HMM separated two groups apart, and the major differences were related to mice's distances to the arena walls (*Figure 5D*).

Finally, we compared the long-range dynamics of these two groups of mice with DTW. DTW results showed a relatively similar dynamic between the two groups, with a minor delay occurring in the control group (*Figure 6—figure supplement 2A*). Inspired by the result of AR-HMM, we wondered if the delay could also be explained by the distance to walls. Therefore, DTW analysis was applied to mice distances to the arena walls, and the result appeared similar (*Figure 6—figure supplement 2B*). When delays of the control group were isolated, we found results generated from Selfee embeddings were strongly consistent with those from mice positions (*Figure 6C*). Although the observed delay

was related to mice positions in the arena, it was only several seconds, and the biological significance was unclear. Further experiments were required to determine whether this result indeed revealed the behavioral dynamic of mice with CIS experiences, or just a deviation due to small sample sizes. In conclusion, our pipeline with Selfee and downstream analysis could evaluate animal behaviors in a purely unsupervised way. The method requires no human definition of either animal behaviors, or key points of animal skeletons, and thus subjective factors that would bias the analysis are avoided.

## Discussion

Here, we use cutting-edge self-supervised learning methods and CNNs to extract Meta-representations from animal behavior videos. Siamese CNNs have proven their capability to learn comprehensive representations (*Chen et al., 2020*). The cosine similarity, part of its loss function used for training, is rational and well suited to measure similarities between the raw images. Besides, CNNs are trained end-to-end so that preprocessing steps like segmentation or key points extraction is unnecessary. By incorporating Selfee with different post-processing methods, we can identify phenotypes of animal behaviors at different time scales. In the current work, we demonstrate that the extracted representations could be used not only for straightforward distance-based analyses such as t-SNE visualization or *k*-NN anomaly detection but also for sophisticated post-processing methods like AR-HMM. These validations confirm that the extracted Meta-representations are meaningful and valuable. Besides anomaly detection, AR-HMM, and DTW discussed here, other methods could also be used to process features produced by Selfee. For classification, as mentioned before, temporal features could be engineered, such as bout features used in FlyTracker, or window features used in JAABA. Also, other unsupervised learning methods developed for skeleton features can be used downstream of Selfee. For example, UMAP non-linear transformations (*McInnes et al., 2018*) and HDBSCAN (*Campello et al., 2013*) clustering proposed in B-SOiD (*Hsu and Yttri, 2021*) and dynamic time alignment kernel proposed in Behavior Atlas (*Huang et al., 2021*). Therefore, Selfee features could be used flexibly with different machine learning methods for various experimental purposes.

By applying our method to mice mating behavior and fly courtship behaviors, we show that Selfee could serve as a helpful complement of the widely used pose estimation methods in multi-animal behavior analysis, and vice versa. First, Selfee features are proved to be comparably discriminative as key points derived human-engineered features. Second, the famous DeepLabCut (*Lauer et al., 2021*; *Mathis et al., 2018*) and similar methods face problems coping with animals of the same color recorded at a compromised resolution and with intensive body contacts. We found that the current version of DeepLabCut could hardly extract useful features during intromission behaviors of two black mice (*Figure 1—figure supplement 3A*, *Figure 1—video 1*). The reason was that it was extremely difficult to unambiguously label body parts like nose, ears and hips when two mice were close enough, a task challenging even for human experts. Similar results were also observed in fly videos. We visualized tracking results provided by the Fly-vs-Fly dataset (*Fleet et al., 2014*), which used classic computer vision techniques for tracking, and inaccurate tracking of fly wings and fly bodies was observed (*Figure 1—figure supplement 3B*, *Figure 1—video 2*). To investigate if cutting-edge deep learning methods would avoid such problems, we labeled 3000 frames to train a SLEAP (*Pereira et al., 2022*) model and inferred on a clip of a fly courtship video. Unfortunately, the same type of error occurred when flies showed intensive body contact (*Figure 1—figure supplement 3C*, *Figure 1—video 3*). Those wrongly tracked frames were either hard for humans to detect or rare postures that were not covered in the training data. By testing these three representative pose estimation programs, we argue that key point detection for closely interacting or overlapped animals is still challenging. By contrast, our methods could capture global characteristics of behaviors like human perception, making it robust to these confusing occlusions. Nevertheless, Selfee features appear less explainable than key points, and we have shown that animal tracking was very useful for interpreting Selfee features. Therefore, Selfee strongly complements the incapability of pose estimation methods processing closely contacted animals, and could be further improved to be more explainable when assisted by animal tracking.

We also demonstrate that the cutting-edge self-supervised learning model is accessible to biology labs. Modern self-supervised learning neural networks usually require at least eight modern GPUs (*Caron et al., 2021*; *Wang et al., 2021*) even TPUs (*Chen et al., 2020*; *Grill et al., 2020*) for training, take advantage of batch sizes larger than 1024, and training time varies from several days to a week.

In contrast, our model can be trained on only one RTX 3090 GPU with a batch size of 256 within only 8 hr with the help of the newly proposed CLD loss function (*Wang et al., 2021*) and other improvements (see Materials and methods for further details). Despite our effort to improve training speed and lower hardware requirement, the current development of self-supervised learning could not make training as accessible as cutting-edge supervised key points detection networks. Nevertheless, we found that our model could achieve zero-shot domain transfer. We demonstrated that Selfee trained for mating behavior of a pair of mice could also be applied to OFTs of single animal. Assisted by AR-HMM, Selfee captured the major differences between mice in the control and CIS groups. Even though the network was trained in a translation-invariant way, and apparent landmarks in the arena were removed during preprocessing, Selfee still identified two distinct behaviors related to mice position in the arena in a zero-shot way. Furthermore, when the model pre-trained with mice videos was applied to rat behaviors, we were able to achieve a zero-shot classification of five major types of social behaviors (*Figure 3—figure supplement 4*). Although the $F_1$ score was only 49.6%, it still captured the major differences between similar behaviors, such as allogrooming and social nose contact (*Figure 3—figure supplement 4*). These results showed that Selfee could be used in a zero-shot training-free way even without GPUs. Thus, we have demonstrated that self-supervised learning could be easily achieved with limited computation resources and a much shorter time and could be directly transferred to datasets that share similar visual characteristics and save more resources.

Despite those advantages, there are some limitations of Selfee. First, because each live-frame only contains three raw frames, our model could not capture much information on the animal motion. It becomes more evident when Selfee is applied to highly dynamic behaviors such as mice mating behaviors. This can be overcome with updated hardware because 3D convolution (*Ji et al., 2013*) or spatial-temporal attention (*Aksan et al., 2020*) is good at dynamic information extraction but requires much more computational resources. Second, as previously reported, CNNs are highly vulnerable to image texture (*Geirhos et al., 2019*). We observed that certain types of beddings of the behavior chamber could profoundly affect the performance of our neural networks (*Figure 1—figure supplement 1*), so in some cases, background removal is necessary (see Materials and methods for further details). Lastly, Selfee could only use discriminative features within each batch, without any negative samples provided, so minor irrelevant differences could be amplified and cause inconsistent results (named mode-split). This mode-split may increase variations of downstream analyses. One possible solution is using some labels to fine-tune the network (cite instance learning). However, the fine-tuning would break the fully unsupervised setup. Another solution is to make the representations more explainable, so that the causes of mode-split can be spotted and corrected. We propose that by combining unsupervised semantic segmentation (*Cho et al., 2021*; *Hamilton et al., 2022*; *Xu et al., 2022*) and bag-of-features (BagNet) (*Brendel and Bethge, 2019*) training objective, it is possible to produce disentangle features that could map each dimension to pixels. Therefore, whenever mode-split happens and corresponding dimensions are identified, and human interference to either recoding setup or training hyperparameters can be applied.

We can envision at least two possible future directions for Selfee. One is to optimize our designs of self-supervised learning method. On the one hand, advanced self-supervised learning methods like DINO (*Caron et al., 2021*) (with visual transformers, ViTs) could separate objects from the background and extract more explainable representations. Besides, by using ViTs, the neural network could be more robust against distractive textures (*Naseer et al., 2021*). At the same time, more temporal information can also be incorporated for a better understanding of motions. Combining these two, equipping ViTs with spatial-temporal attention could extract better features. On the other hand, although Siamese networks are popular choices for self-supervised learning, they require two times more computational resources than single-branch designs. A recent work on instance learning shed light on self-supervised learning with smaller datasets and simpler architectures. This could be a promising direction in which self-supervised learning for animal behaviors could be more accessible for biologists. In summary, possible improvements of Selfee designs can either bring in more advanced and complex architectures for better performance or try more simplified instance learning techniques to achieve easier deployment.

Another direction will be explainable behavior forecasting for a deeper understanding of animal behaviors. For a long time, behavior forecasting has been a field with extensive investigations in which RNNs, LSTMs, or transformers are usually applied (*Aksan et al., 2020*; *Fragkiadaki, 2015*;

*Sun et al., 2021*). However, most of these works use coordinates of key points as inputs. Therefore, the trained model might predominantly focus on spatial movement information and discover fewer behavioral syntaxes. By representation learning, spatial information is essentially condensed so that more syntaxes might be highlighted. Transformer models for forecasting could capture correlations between sub-series as well as long-term trends like seasonality (*Wu et al., 2021*). These deep learning methods would provide behavioral neuroscientists with powerful tools to identify behavior motifs and syntaxes that organize stereotyped motifs beyond the Markov property.

# Materials and methods

## Key resources table

| Reagent type (species) or resource | Designation | Source or reference | Identifiers | Additional information |
|---|---|---|---|---|
| Genetic reagent (*Drosophila melanogaster*) | $w^{1118}$ | – | – | Female, **Figure 3A–C** & **Figure 5F–G**; male, **Figure 4G–J** |
| Genetic reagent (*Drosophila melanogaster*) | CS | – | – | Male, **Figure 3A–C**, **Figure 4B** & **Figure 5F–G** |
| Genetic reagent (*Drosophila. melanogaster*) | $CCHa1^{attP}$ | BDRC | 84458 | w[*]; TI{RFP[3xP3.cUa]=TI}CCHa1[attP]; male, **Figure 4B** |
| Genetic reagent (*Drosophila melanogaster*) | $CCHa1\text{-}R^{attP}$ | BDRC | 84459 | w[*]; TI{RFP[3xP3.cUa]=TI}CCHa1-R[attP]; male, **Figure 4B** |
| Genetic reagent (*Drosophila melanogaster*) | $CCHa2^{attP}$ | BDRC | 84460 | w[*]; TI{RFP[3xP3.cUa]=TI}CCHa2[attP]; male, **Figure 4B** |
| Genetic reagent (*Drosophila melanogaster*) | $CCHa2\text{-}R^{attP}$ | BDRC | 84461 | w[*]; TI{RFP[3xP3.cUa]=TI}CCHa2-R[attP]; male, **Figure 4B** |
| Genetic reagent (*Drosophila melanogaster*) | $CCHa2\text{-}R\text{-}RA^{Gal4}$ | BDRC | 84603 | TI{2 A-GAL4}CCHa2-R[2 A-A.GAL4]; with Kir2.1, **Figure 4B** |
| Genetic reagent (*Drosophila melanogaster*) | $CCHa2\text{-}R\text{-}RB^{Gal4}$ | BDRC | 84604 | TI{2 A-GAL4}CCHa2-R[2A-B.GAL4]; with Kir2.1, **Figure 4B** |
| Genetic reagent (*Drosophila melanogaster*) | $CNMa^{attP}$ | BDRC | 84485 | w[*]; TI{RFP[3xP3.cUa]=TI}CNMa[attP]; male, **Figure 4B** |
| Genetic reagent (*Drosophila melanogaster*) | $Oamb^{attP}$ | BDRC | 84555 | w[*]; TI{RFP[3xP3.cUa]=TI}Oamb[attP]; male, **Figure 4B** |
| Genetic reagent (*Drosophila melanogaster*) | $Dop2R^{KO}$ | BDRC | 84720 | TI{TI}Dop2R[KO]; male, **Figure 4B** |
| Genetic reagent (*Drosophila melanogaster*) | $DopEcR^{Gal4}$ | BDRC | 84717 | TI{GAL4}DopEcR[KOGal4.w-]; male, **Figure 4B** |
| Genetic reagent (*Drosophila melanogaster*) | $SerT^{attP}$ | BDRC | 84572 | w[*]; TI{RFP[3xP3.cUa]=TI}SerT[attP]; male, **Figure 4B** |
| Genetic reagent (*Drosophila melanogaster*) | $Trh^{Gal4}$ | BDRC | 86146 | w[*]; TI{RFP[3xP3.cUa]=2 A-GAL4}Trh[GKO]; male, **Figure 4B** |
| Genetic reagent (*Drosophila melanogaster*) | $TK^{attP}$ | BDRC | 84579 | w[*]; TI{RFP[3xP3.cUa]=TI}Tk[attP]; male, **Figure 4B** |
| Genetic reagent (*Drosophila melanogaster*) | UAS-Kir2.1 | BDRC | 6595 | w[*]; P{w[+mC]=UAS-Hsap\KCNJ2.EGFP}7; with Gal4, **Figure 4B** |
| Genetic reagent (*Drosophila melanogaster*) | $NorpA^{36}$ | BDRC | 9048 | w[*] norpA[P24]; male, **Figure 5F–G** |
| Genetic reagent (*Drosophila melanogaster*) | $Tdc2^{RO54}$ | Pan Lab at SEU | | Tdc2[RO54]; male, **Figure 4B** |
| Genetic reagent (*Drosophila melanogaster*) | Taotie-Gal4 | Zhu Lab at IBP | | w[*]; P{w[+mC]=Gr28 b.b-GAL4.4.7}10; with Kir2.1, **Figure 4B** |
| Genetic reagent (*Mus musculus*) | C57BL/6J | – | – | **Figure 3D–F** & **Figure 5B–D** |
| Software, algorithm | python | Anaconda | – | 3.8.8 |
| Software, algorithm | numpy | Anaconda | – | 1.19.2 |
| Software, algorithm | matplotlib | Anaconda | – | 3.4.1 |

*Continued on next page*

*Continued*

| Reagent type (species) or resource | Designation | Source or reference | Identifiers | Additional information |
|---|---|---|---|---|
| Software, algorithm | av | conda-forge | – | 8.0.3 |
| Software, algorithm | scipy | Anaconda | – | 1.6.2 |
| Software, algorithm | cudatoolkit | conda-forge | – | 11.1.1 |
| Software, algorithm | pytorch | pytorch | – | 1.8.1 |
| Software, algorithm | torchvision | pytorch | – | 0.9.1 |
| Software, algorithm | pillow | Anaconda | – | 8.2.0 |
| Software, algorithm | scikit-learn | Anaconda | – | 0.24.2 |
| Software, algorithm | pandas | Anaconda | – | 1.2.4 |
| Software, algorithm | lightgbm | conda-forge | – | 3.2.1 |
| Software, algorithm | opencv-python | PyPI | – | 4.5.3.56 |
| Software, algorithm | psutil | PyPI | – | 5.8.0 |
| Software, algorithm | pytorch-metric-learning | PyPI | – | 0.9.99 |
| Software, algorithm | pyhsmm | PyPI | – | 0.1.6 |
| Software, algorithm | autoregressive | PyPI | – | 0.1.2 |
| Software, algorithm | dtw-python | PyPI | – | 1.1.10 |
| Software, algorithm | SLEAP | conda-forge | – | 1.2.2 |
| Software, algorithm | DEEPLABCUT | PyPI | – | 2.2.0.2 |

## Fly stocks

All fly strains were maintained under a 12 hr/12 hr light/dark cycle at 25°C and 60% humidity (PERCIVAL incubator). The following fly lines were acquired from Bloomington *Drosophila* Stock Center: *CCHa1*$^{attP}$ (84458), *CCHa1-R*$^{attP}$ (84459), *CCHa2*$^{attP}$ (84460), *CCHa2-R*$^{attP}$ (84461), CCHa2-R-RA$^{Gal4}$ (84603), CCHa2-R-RB$^{Gal4}$ (84604), *CNMa*$^{attP}$ (84485), *Oamb*$^{attP}$ (84555), *Dop2R*$^{KO}$ (84720), *DopEcR*$^{Gal4}$ (84717), *SerT*$^{attP}$ (84572), *Trh*$^{Gal4}$ (86146), *TK*$^{attP}$ (84579), *NorpA*$^{36}$ (9048), UAS-Kir2.1 (6595). *Tdc2*$^{RO54}$ was a gift from Dr Yufeng Pan at Southeast University, China. Taotie-Gal4 was a gift from Dr Yan Zhu at Institute of Biophysics, Chinese Academy of Sciences, China.

## Fly courtship behavior and male-male interaction

Virgin female flies were raised for 4–6 days in 15-fly groups, and naïve male flies were kept in isolated vials for 8–12 days. All behavioral experiments were done under 25°C and 45–50% humidity. Flies were transferred into a customized chamber of 3 mm in height and 10 mm in diameter by a home-made aspirator. Fly behaviors were recorded using a stereoscopic microscope mounted with a CCD camera (Basler ORBIS OY-A622f-DC) at the resolution of 1000×500 (for two chambers at the same time), or 640×480 (for individual chambers) and a frame rate of 30 Hz. Five types of behaviors were annotated manually, including 'chasing' (a male fly follows a female fly), 'wing extension' a male fly extends unilateral wing and orientates to the female to sing courtship son, 'copulation attempt' (a male fly bends its abdomen toward the genitalia of the female or the unstable state that male fly mounts on a female with its wings open), and 'copulation' (male fly mounts on a female in a stable posture for several minutes).

## Fly defensive behavior assay

The kicking behavior was tested based on previously reported paradigms (*Li et al., 2016*; *Liu et al., 2020*). Briefly, flies were raised in groups for 3–5 days. Flies were anesthetized on ice, and then male flies were decapitated and transferred to 35 mm Petri dishes with damped filter paper on the bottom to keep the moisture. Flies were allowed to recover for around 30 min in the dishes. The probe for stimulation was homemade from a heat-melt yellow pipette tip, and the probe's tip was 0.3 mm.

Each side of flies' wing margin was gently touched five times, and the kicking behavior was recorded manually. The statistical analysis was performed with the Mann-Whitney test with GraphPad Prism Software.

## Social behavior assay for flies

The social distance was tested based on the previously reported method (*McNeil et al., 2015*). Briefly, flies were raised in groups for 3 days. Flies were anesthetized paralyzed on ice, and male flies were picked and transferred to new vials (around 40 flies per vial). Flies were allowed to recover for 1 day. The vertical triangular chambers were cleaned with 75% ethanol and dried with paper towels. After assembly, flies were transferred into the chambers by a homemade aspirator. The photos were taken after 20 min, and the positions of each fly were manually marked in ImageJ. The social distances were measured with the lateral sides of the chambers (16.72 cm) as references, and the median values of the nearest, median, and average distance of each replication are calculated. The statistical analysis was performed with the Mann-Whitney test in GraphPad Prism Software.

## Mice mating behavior assay

Wild-type mice of C57BL/6J were purchased from Slac Laboratory Animal (Shanghai). Adult (8–24 weeks of age) male mice were used for sexual behavior analysis. All animals were housed under a reversed 12 hr/12 hr light-dark cycle with water and food ad libitum in the animal facility at the Institute of Neuroscience, Shanghai, China. All experiments were approved by the Animal Care and Use Committee of the Institute of Neuroscience, Chinese Academy of Sciences, Shanghai, China (IACUC No. NA-016-2016).

Male mice were singly housed for at least 3 days prior to sexual behavioral tests. All tests were initiated at least 1 hr after lights were switched off. Behavioral assays were recorded using infrared cameras at the frame rate of 30 Hz. Female mice were surgically ovariectomized and supplemented with hormones to induce receptivity. Hormones were suspended in sterile sunflower seed oil (Sigma-Aldrich, S5007) and injected 10 mg (in 50 mL oil) and 5 mg (in 50 mL oil) of 17b-estradiol benzoate (Sigma-Aldrich, E8875) 48 and 24 hr preceding the test, respectively. On the day of the test, 50 mg of progesterone (Sigma-Aldrich, P0130; in 50 mL oil) was injected 4–6 hr prior to the test. Male animals were adapted 10 min to behavioral testing rooms where a recording chamber equipped with video acquisition systems was located. A hormonal primed ovariectomized C57BL/6J female (OVX) was introduced to the home cage of male mice and videotaped for 30 min. Mating behavior tests were repeated three times with different OVX at least 3 days apart. Videos were manually scored using a custom-written MATLAB program. The following criteria were used for behavioral annotation: active nose contacts initiated by male mouse toward the female's genitals, body area, faces were defined collectively as 'social interest'; male mouse climbs the back of the female and moves the pelvis were defined as 'mount'; rhythmic pelvic movements after mount were defined as 'intromission'; a body rigidity posture after final deep thrust were defined as 'ejaculation'.

## Mice OFT

All experiments were performed using the principles outlined in *the Guide for the Care and Use of Laboratory Animals of Tsinghua University*. C57BL/6J male mice, aged 8–12 weeks, were used for behavior test. Mice were housed five per cage with free access to food and water and under a 12 hr light-dark cycle (light on from 7 p.m. to 7 a.m.). All mice were purchased and maintained under standard conditions by the Animal Research Centre of Tsinghua University.

All studies and experimental protocols were approved by Institutional Animal Care and Use Committee (IACUC) at Tsinghua University (No. 19-ZY1). Specifically, the OFT was conducted in an open plastic arena (50 cm × 50 cm × 40 cm). Mice were first placed in the peripheral area with their head toward the wall. Exploration time during 10 min in the peripheral and central regions, respectively, were measured using an automated animal tracking program.

The animal tracking program was coded in Python 3 with Open-CV. Each frame was first preprocessed, and then a median filter with a kernel size of 5 and a threshold of 150 was performed sequentially. Connected components with the maximum area were extracted, and the center of gravities and borders were visualized with original images for manual proofreading. Tracking results were saved in plain text format.

## Data preprocessing, augmentation, and sampling

Fly behavior videos were decomposed into frames by FFmpeg, and only the first 10,000 frames of each video were preserved and resized into images with a resolution of 224×224. For model training of *Drosophila* courtship behavior, each video was manually checked to ensure successful copulations within 10,000 frames.

Mice behavior videos were decomposed into frames by FFmpeg, and only frames of the first 30 min of each video were preserved. Frames were then preprocessed with OpenCV (*Bradski, 2000*) in Python. Behavior chambers in each video were manually marked, segmented, and resized into images of a resolution of 256 × 192 (mating behavior) or 500 × 500 (OFT). For background removal, the average frame of each video was subtracted from each frame, and noises were removed by a threshold of 25 and the median filter with a kernel size of 5. Finally, the contrast was adjusted with histogram equalization.

For data augmentations, crop, rotation, flip, Turbo, and color jitter were applied. For a given frame, it formed a live-frame with its preceding and succeeding frames. For flies' behavior video, three frames were successive, and for mice, the preceding or succeeding frame is one frame away from the current frame due to their slower dynamics (*Wiltschko et al., 2015*). Each live-frame was randomly cropped into a smaller version containing more than 49% (70%×70%) of the original image; then the image was randomly (clockwise or anticlockwise) rotated for an angle smaller than the acute angle formed by the diagonal line and the vertical line, then the image would be vertically flipped, horizontally flipped, and/or applied the Turbo lookup table (*Mikhailov, 2019*) at the probability of 50%, respectively; and finally, the brightness, contrast, saturation, and hue were randomly adjusted within 10% variation. Notably, since the Turbo transformation is designed for grayscale images, for a motion-colored RGB image, each channel was transformed individually. After Turbo transformation, their corresponded channels were composited to form a new image.

For fly data sampling, all images of all videos were randomly ranked, and each batch contained 256 images from different videos. For mice data sampling, all images of each video were randomly ranked, and each batch contained 256 images from the same video. This strategy was designed to eliminate the inconsistency of recording conditions of mice that was more severe than flies.

## Selfee neural network and its training

All training and inference were accomplished on a workstation with 128 GB RAM, AMD Ryzen 7 5800×, and one NVIDIA GeForce RTX 3090. Selfee neural network was constructed based on publications and source codes of BYOL (*Grill et al., 2020*), SimSiam (*Chen et al., 2020*), and CLD (*Wang et al., 2021*) with PyTorch (*Paszke et al., 2019*). In brief, the last layer of ResNet-50 was removed, and a three-layer 2048-dimension MLP was added as the projector. Hidden layers of the projector were followed by batch normalization (BN) and ReLU activation, and the output layer only had BN. The predictor was constructed with a two-layer bottleneck MLP with a 512-dimension hidden layer and a 2048-dimension output layer. The hidden layer but not the output layer of the predictor had BN and ReLU. As for the group discriminator for CLD loss, it had only one normalized fully connected layer that projected 2048-dimension output to 1024 dimensions, followed by a customized normalization layer that was described in the paper of CLD (*Wang et al., 2021*). The collapse level was monitored as one minus to average standard deviation of each channel of the normalized representation multiplied by the square root of the channel number. If a collapse happens, the standard deviation becomes zero, and the collapse level should be one. If no collapse happens, each channel should obey standard normal distribution, and the average standard deviation is one. The normalization operation cancels out the square root of the channel number. In this way, the collapse level is zero.

The loss function of Selfee had two major parts. The first part was the negative cosine loss (*Chen et al., 2020*; *Grill et al., 2020*), and the second part was the CLD loss (*Wang et al., 2021*). For a batch of $n$ samples, $Z$, $P$, $V$ represented the output of projector, predictor, and group discriminator of the main branch, respectively; $Z'$, $P'$, $V'$ represented the output of the reference branch; and $sg$ as the stop-gradient operator. After $k$-means clustering of $V$, the centroids of $k$ classes were given by $M$, and labels of each sample were provided in the one-hot form as $L$. The hyperparameter $\theta$ was 0.07, and $\lambda$ was 2. The loss function was given by the following equations:

$$\text{CosineDistance}(m, n) = 1 - \frac{m}{\|m\|_2} \cdot \frac{n}{\|n\|_2}$$

$$\text{Loss}_1 = \frac{1}{2n}\sum_{i=1}^{n}\text{CosineDistance}(sg(z_i), p'_i) + \frac{1}{2n}\sum_{i=1}^{n}\text{CosineDistance}(sg(z'_i), p_i)$$

$$\text{CrossEntropyLoss}(x, l) = -x.l + \log(\| \exp(x) \|_1)$$

$$\text{Loss}_2 = \frac{1}{2}\text{CrossEntropyLoss}\left(\frac{\nu_i}{\theta}.M'^T, l_i\right) + \frac{1}{2}\text{CrossEntropyLoss}\left(\frac{\nu'_i}{\theta}.M^T, l'_i\right)$$

$$\text{Loss} = \text{Loss}_1 + \lambda\,\text{Loss}_2$$

For all training processes, the Selfee network was trained for 20,000 steps with the SDG optimizer with a momentum of 0.9 and a weight decay of 1e-4. The learning rate was adjusted in the one-cycle learning rate policy (*Smith and Topin, 2017*) with base learning rates and a pct start of 0.025. The model for *Drosophila* courtship behavior was initialized with ResNet-50 pre-trained on the ImageNet, and the base learning rate was 0.025 per batch size of 256. As for the mating behaviors of mice, the model was initialized with weights trained on the fly dataset, and the base learning rate was 0.05 per batch size of 256.

For fly courtship behavior, 516 and 7 videos (4,607,274 and 55,708 frames) were used as train set and test set, respectively; for mice mating behavior, 118 and 13 videos (4,943,101 and 548,993 frames) were used as train set and test set, respectively. For comparison with animal tracking methods, Selfee was fine-tuned with 11 and 1 videos (1,188,550 and 108,050 frames).

## t-SNE visualization

Video frames for t-SNE visualization were all processed by Selfee. Embeddings of three tandem frames were averaged to eliminate potential noises. All embeddings were transformed using t-SNE provided in the scikit-learn (*Paszke et al., 2019*) package in Python without further tuning of parameters. Results were visualized with the Matplotlib (*Hunter, 2007*) package in Python, and their colors were assigned based on human annotations of video frames.

## Classification

Two kinds of classification methods were implemented, including the *k*-NN classifier and the LightGBM classifier. The weighed *k*-NN classifier was constructed based on the previous reports (*Chen et al., 2020*; *Wu et al., 2018*). LightGBM classifier (*Ke et al., 2017*) was provided by its official package in Python. The $F_1$ score and mAP were calculated with the scikit-learn (*Paszke et al., 2019*) package in Python. Scores for different type of behaviors were averaged in the 'macro' way, while scores for behaviors of different intensity were averaged in the 'micro' way.

For fly behavior classification, seven 10,000-frame videos were annotated manually. Sevenfold cross-validation was performed using embeddings generated by Selfee and the *k*-NN classifier. Inferred labels were forced to be continuous through time using inferred labels of 21 neighbor frames to determine the final result. The neighborhood length was determined by plotting the similarity between neighbor frames and the current frame and choosing the frame number where similarities dropped to the half between the peak and the bottom. Then, a video independent of the cross-validation was annotated and inferred by a *k*-NN classifier using all 70,000 samples, and the last 3000 frames were used for the raster plot.

To compare different types of features, three published labeled videos were used. Each frame of video was labelled with a wing extension score from 0 to 6 (scored from 0 to 3 by two individuals). Each video of 30 min was evenly split into two parts, and six video samples were used for sixfold cross-validation. Voting by 21 neighbors was not used. FlyTracker features and JAABA features were obtained from the original publication (https://library.ucsd.edu/dc/object/bb20197654), and distances between key points were calculated from the tracking results of FlyTracker.

For rat behavior classification, the RatSI dataset (*Lorbach et al., 2018*) (a kind gift from Noldus Information Technology bv) contains nine manually annotated videos. We neglected three rarest annotated behaviors: moving away, nape attacking, and pinning, and we combined approaching and following into a larger category. Therefore, we used five kinds of behaviors, including allogrooming, approaching or following, social nose contact, solitary, and others. Ninefold cross-validation was performed using embeddings generated by Selfee and the *k*-NN classifier. Inferred labels were forced to be continuous through time by using inferred labels of 81 neighbor frames to determine the final result.

For mice behavior classification, eight videos were annotated manually. Eightfold cross-validation was performed using embeddings generated by Selfee and the *k*-NN classifier. To incorporate more temporal information, the LightGBM classifier and additional features were also used. Additional features include slide moving average and standard deviation of 81-frame time windows, the main frequencies, and their energy (using short-time Fourier transform in SciPy; *Virtanen et al., 2020*) within 81-frame time windows. Early-stop was used to prevent over-fitting. Inferred labels were forced to be continuous through time by using inferred labels of 81 neighbor frames to determine the final result. Then, a video independent of the cross-validation was annotated and inferred by an ensemble classifier of eight previously constructed classifiers, and all frames were used for the raster plot.

## Anomaly detection

For a group of query embeddings of sequential frames $q_1$, $q_2$, $q_3$, …, $q_n$, and a group of reference embeddings of sequential frames $r_1$, $r_2$, $r_3$, …, $r_m$, the anomaly score of each query frame was given by the following equation:

$$\text{AnomalyScore}(q_i) = \min_{j=1}^{m} \left( \text{CosineDistance}(q_i, r_j) \right) - \min_{|j-i| < 50} \left( \text{CosineDistance}(q_i, q_j) \right)$$

A PyTorch implementation of cosine similarity (*Musgrave et al., 2020*) was used for accelerated calculations.

The anomaly score of each video was the average anomaly score of the top 100 anomalous frames. The statistical analysis of the genetic screening was performed with one-way ANOVA with Benjamini and Hochberg correction in GraphPad Prism Software.

If negative controls are provided, anomalous frames are defined as frames with higher anomaly scores than the maximum anomaly score of frames in negative control videos.

## Autoregressive hidden Markov model

All AR-HMMs were built with the implementation of MoSeq (*Wiltschko et al., 2015*) (https://github.com/mattjj/pyhsmm-autoregressive; *Linderman, 2018*). A PCA model that could explain 95% of variance of the control group was built and used to transform both control and experiment groups. The max module number was set as 10 for all experiments unless indicated otherwise. Each model was sampled for 1000 iterations. We kept other hyperparameters the same as the examples provided by this package. State usages of each module in control and experimental groups were analyzed by Mann-Whitney test with SciPy *Virtanen et al., 2020* followed with Benjamini and Hochberg correction. The state usages were also visualized after PCA dimensional reduction with scikit-learn (*Paszke et al., 2019*) and Matplotlib (*Hunter, 2007*) for the exclusion of possible obvious outliners or batch effects.

## Dynamic time warping

DTW was modified from the Python implementation (*Toni, 2009*) (https://dynamictimewarping.github.io/python/). Specifically, PyTorch implementation of cosine similarity (*Musgrave et al., 2020*) was used for accelerated calculations.

## Pose estimation with SLEAP

We used the official implementation of SLEAP. For explanation for t-SNE plot of Selfee features, we labeled the same 3000 frames with the help of SLEAP. First, we labeled around 400 frames and trained a SLEAP model. The trained SLEAP model was used to infer all 3000 frames, tracking was performed using the 'simple' method, and the target number of instances per frame was set to 2. Inferred skeletons and tracking results were manually proofread and corrected. These results were used for the following analysis of the t-SNE plot. Then, all 3000 frames were used to train a new SLEAP model. This new model was used for the pose estimation of another clip of a fly courtship video (shown in *Figure 1—videos 1–3* and *Figure 1—figure supplement 3*).

## Visualization tracking results from FlyTracker

Tracking results of the Fly-vs-Fly dataset were obtained from its official website. The head and tail coordinates were calculated from the center, the orientation, and the major axis length. Tracking results were visualized with OpenCV.

## Pose estimation with DeepLabCut

We used the official implementation of DeepLabCut (*Lauer et al., 2021*; *Mathis et al., 2018*). For training, 120 frames of a mating behavior video were labeled manually, and 85% were used as the training set. Marked body parts included nose, ears, body center, hips, and bottom, following previous publications (*Segalin et al., 2020*; *Sun et al., 2021*). The model (ResNet-50 as the backbone) was trained for 100,000 iterations, with a batch size of 16. We kept other hyperparameters the same as default settings.

## Data and code availability statement

Major datasets that support the findings of this study are available from Dryad (https://doi.org/10.5061/dryad.brv15dvb8). Other data are available from the corresponding author upon reasonable request, because they are too large to upload to a particular server. Source codes used in this article are available on GitHub (https://github.com/EBGU/Selfee; *Jia, 2022*; copy archived at swh:1:rev:3af2d1ed2dfcf3bd1d1c18d488ed87f7b826529c).

## Acknowledgements

We thank members of the Zhang lab for discussions. This work was supported by grants 31871059 and 32022029 from the National Natural Science Foundation of China, grant Z181100001518001 from the Beijing Municipal Science & Technology Commission, and a 'Brain + X' seed grant from the IDG/McGovern Institute for Brain Research at Tsinghua to WZ. WZ is supported by Chinese Institute for Brain Research, Beijing. WZ is an awardee of the Young Thousand Talent Program of China.

## Additional information

### Funding

| Funder | Grant reference number | Author |
| --- | --- | --- |
| National Natural Science Foundation of China | 32022029 | Wei Zhang |
| National Natural Science Foundation of China | 31871059 | Wei Zhang |
| Beijing Municipal Science and Technology Commission | Z181100001518001 | Wei Zhang |
| Tsinghua University | IDG/McGovern Institute for Brain Research - Brain + X | Wei Zhang |
| Chinese Institute for Brain Research, Beijing | | Wei Zhang |
| Young Thousand Talent Program of China | | Wei Zhang |

The funders had no role in study design, data collection and interpretation, or the decision to submit the work for publication.

### Author contributions

Yinjun Jia, Conceptualization, Data curation, Formal analysis, Investigation, Resources, Software, Writing - original draft, Writing – review and editing; Shuaishuai Li, Resources, Writing – review and editing; Xuan Guo, Data curation, Resources; Bo Lei, Resources; Junqiang Hu, Data curation, Software; Xiao-Hong Xu, Supervision, Writing – review and editing; Wei Zhang, Conceptualization, Project administration, Supervision, Writing - original draft, Writing – review and editing

### Author ORCIDs

Yinjun Jia (iD) http://orcid.org/0000-0002-8281-0669
Wei Zhang (iD) http://orcid.org/0000-0003-0512-3096

### Ethics

All mating experiments were approved by the Animal Care and Use Committee of the Institute of Neuroscience, CAS Center for Excellence in Brain Science and Intelligence Technology, Chinese Academy of Sciences, Shanghai, China (IACUC No. NA-016-2016) All studies and experimental protocols of CIS and OFT were approved by Institutional Animal Care and Use Committee (IACUC) at Tsinghua University (No. 19-ZY1). Experiments were performed using the principles outlined in the Guide for the Care and Use of Laboratory Animals of Tsinghua University.

### Decision letter and Author response

Decision letter https://doi.org/10.7554/eLife.76218.sa1
Author response https://doi.org/10.7554/eLife.76218.sa2

## Additional files

### Supplementary files

• MDAR checklist

### Data availability

Major data used in this study were uploaded to Dryad, including pretrained weights. Data can be accessed via: https://doi.org/10.5061/dryad.brv15dvb8. With the uploaded dataset and pretrained weights, our experiments can be replicated. However, due to its huge size and the limited internet service resources, we are currently not able to share our full training dataset. The full dataset is as large as 400GB, which is hard to upload to a public server and will be difficult for others users to download. The training dataset, is available from the corresponding author upon reasonable request (https://www.ie.tsinghua.edu.cn/eng/info/1017/1347.htm), and then we can discuss how to transfer the dataset. No project proposal is needed as long as the dataset is not used for any commercial purpose. Our Python scripts can be accessed on GitHub: https://github.com/EBGU/Selfee, copy archived at swh:1:rev:3af2d1ed2dfcf3bd1d1c18d488ed87f7b826529c. Other software used in our project include ImageJ (https://imagej.net/software/fiji/) and GraphPad Prism (https://www.graphpad.com/). All data used to plot graphs and charts in the manuscript can be fully accessed on Dryad (DOI 10.5061/dryad.brv15dvb8).

The following dataset was generated:

| Author(s) | Year | Dataset title | Dataset URL | Database and Identifier |
|---|---|---|---|---|
| Jia Y, Li S, Guo X, Lei B, Hu J, Xu X, Zhang W | 2022 | Data from: Selfee: Self-supervised features extraction of animal behaviors | https://dx.doi.org/10.5061/dryad.brv15dvb8 | Dryad Digital Repository, 10.5061/dryad.brv15dvb8 |

The following previously published datasets were used:

| Author(s) | Year | Dataset title | Dataset URL | Database and Identifier |
|---|---|---|---|---|
| Eyrun E, Branson S, Burgos-Artizzu P, Hoopfer ED, Schor J, Anderson DJ, Perona P | 2021 | Fly-vs-Fly | https://data.caltech.edu/records/1893 | CaltechDATA, 1893 |
| Xubo L, Margot W, Kenichi I, Pavan N, Kenta A | 2020 | Data from: Quantifying influence of human choice on the automated detection of *Drosophila* behavior by a supervised machine learning algorithm | https://library.ucsd.edu/dc/object/bb20197654 | LIBRARY DIGITAL COLLECTIONS, 10.6075/J0QF8RDZ |
| Lorbach M, Kyriakou EI, Poppe R, van Dam EA, Noldus LPJJ, Veltkamp RC | 2017 | RatSI | https://www.noldus.com/form/ratsi-dataset | Noldus Information Technology, ratsi-dataset |

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
