## [Editor Report]

Jia et al., present a valuable machine learning framework, Selfee, based on deep neural networks for analyzing video recordings of animal behavior, which is efficient and runs in an unsupervised fashion, and requires very little pre-processing. Selfee should be of broad interest to researchers studying quantitative animal behavior.

---

## [Decision Letter]

**Decision letter after peer review:**

Thank you for submitting your article "Selfee: Self-supervised Features Extraction of animal behaviors" for consideration by *eLife*. Your article has been reviewed by 3 peer reviewers, and the evaluation has been overseen by a Reviewing Editor and K VijayRaghavan as the Senior Editor. The following individuals involved in review of your submission have agreed to reveal their identity: Mason Klein (Reviewer #1); Primoz Ravbar (Reviewer #3).

The Reviewers agree that the paper presents a really exciting method that could have broad impact on researchers studying quantitative animal behavior. However there are some unresolved issues for establishing the credibility of the method that we would like you to address.

Essential revisions:

1) Chose one or two strong and clean datasets and take them all the way through the analysis pipeline (selfee->anomaly score->AR-HMM->DTW) instead of using different experiments for explaining different points. We suggest removing IR76b data from the story unless the experiment can be somehow justified.

2) Also show Selfee's results on a simpler data set (with *Drosophila* or some other animal) with single-animal video.

3) Explain the rationale behind the choice of parameters for "Selfee". Please explain why Selfee performs so poorly without the CLD component. Would a simpler architecture, with the addition of CLD, achieve similar performance as the Selfee?

4) Explore the behaviors found in different parts of the t-SNE maps and evaluate the effect of the irrelevant features on their distributions. The t-SNE maps display several clusters representing the same behaviors (Figure 2D) which suggests strong overfitting (weak generalization). The authors should at least show some examples taken from different clusters. This would illustrate why Selfee is separating behaviors that humans perceive as a single category.

5) Provide a formal comparison with one of the pre-existing methods that would in theory work on the same videos (e.g. JAABA) (Posted 6th Apr 2022).

6) Provide a supplementary table showing full genotype of all experimental animals.

*Reviewer #1 (Recommendations for the authors):*

Overall I like the paper! I will try to give as many specific, detailed comments as possible in this section.

First, to clarify some of the items I listed under "weaknesses" in the public review:

(1) Since the paper feels mainly like a Selfee methods paper, I think spending more time describing what exactly Selfee is and how it works would be appropriate. I would recommend putting an expanded version of the Methods section paragraph (lines 925-933) in the main body of the paper, briefly explaining the techniques and existing software as you go through each step. It would give a reader with little machine learning experience a better understanding of the process.

(2) Efficiency was talked about in the introduction and occasionally brought up when highlighting that Selfee doesn't need large computational/processing power to operate, but I didn't come away with a sense of how much Selfee improves on other approaches in this regard. It would be helpful to be more specific.

(3) The comparisons with DeepLabCut make sense -- if it would be straightforward to compare to other software, that might help highlight unique aspects of Selfee a bit more effectively.

(4) I don't usually recommend new experiments when I review papers, but perhaps showing Selfee's results on a simpler data set (with *Drosophila* or some other animal), say with single-animal video, before showing the courtship-type results, could ease a reader into the process more effectively.

(5) It would also be great, if possible, to include a small section that could operate like a guide for a novice user to try Selfee on their own data.

A few general comments:

(1) I am not a machine learning expert myself, so I hope another reviewer could evaluate Selfee's uniqueness compared to other methods, as I am taking that at face value in my review. In particular, (line 17-19), is this true that unsupervised analysis has not been used this way before?

(2) A fair amount of grammar/word usage errors, not really a problem and I assume copy editors will address? (e.g. Fast  Quickly or Rapidly on line 15).

(3) I generally like the figures, but found Figure 2 and Figure 3 pretty difficult to read at that size. Rearranging the panels and/or using larger fonts would help readability considerably.

*Reviewer #2 (Recommendations for the authors):*

This work is very relevant but has some serious unresolved issues before warranting further consideration. The overall format of the paper will also have to be massively restructured to make it palatable for a wide, life-science research audience.

Major points of concern that need to be addressed:

1) In the way it is presented, the rationale behind choice of several of the parameters for "Selfee" are unclear to me. E.g.

a. Why do the live frames consist of 3 tandem frames? Is 3 somehow a special number that is easy to compute or represent as an RGB image? Why not 4 with CMYK or a higher number with a distinct colormap as is commonly done for representing depth coding.

b. The choice of AR-HMM and DTW as post-processing steps after the Selfee analysis is also a bit unclear to me. Especially because the authors do not discuss any alternatives and moreover, they don't even show great promise in the examples later in the manuscript.

2) The authors have surprisingly failed to acknowledge and cite the state of the field that directly translates to their novelty claims (Lines 56-59: "There are no pose estimation paradigms that are designed to handle multi-animal contexts"). Both multi-animal DLC (Lauer et al., 2021) and SLEAP (Perreira et al., 2020) were explicitly designed for this purpose. Not to mention a plethora tools for multi-animal behavioral classification, especially during social interactions that provide pretty much all the extracted information obtained using Selfee in this manuscript. Listing a few *Drosophila* specific tools:

a. JAABA (Kaabra et al., Nat Methods, 2013)

b. Matebook (Ribeiro et al., Cell 2018)

c. Flytracker (Hoopfer et al., *eLife* 2015)

3) The authors must provide a sup table showing full genotype of all experimental animals. Wherever possible test and controls should be in the same genetic background (this is especially true for loss of function experiments where vision and locomotion is an integral part of the behavior, like the ones done in this manuscript). Legends for videos should also be provided.

4) Figure 2 demonstrates how good Selfee is at extracting information. To do this, authors have to either look at how data clusters with respect to human annotated labels on a t-sne map or use a k-nn classification step to classify the data. Honestly, the t-sne maps in Figure 2D don't look that promising. K-nn classification looks better with around 70% F1 score. But at this point there should be a formal comparison with one of the pre-exisiting methods that would in theory work on the same videos. I suggest JAABA (or Flytracker+JAABA) since similar labels have been previously extracted using this toolkit. Same holds true for mouse data (2G-I) and especially more important because the F1 scores here are a bit low. Also panels 2F and 2I likely represent cherry picked cases where results are much better than what the scores indicate and hence are not really informative.

5) The rationale for definition of "anomaly score" (Figure 3, lines 230-249) is again poorly defended in the text. Could authors clearly define what "meta-representations of frames" means. Also, why restrict k=1 for IES? How is the +-50 frame window chosen for IAS score. Are these empirically derived values after some trial and error, or based on some priors. Since the anomaly score is quite central to the overall utility of this tool, it needs to be proposed in a much more credible way.

6) The unusual subset of neurotransmitter mutants/silenced lines used in Figure 3 data makes one wonder if the provided data is a cherry-picked subset of the full data to make the story easier. I would really appreciate if all data were shown clearly and rationally. The authors really need to establish credibility for persuading the reader to use their method.

7) Trh mutant phenotype is interesting but interpretations of secondary assays warrant some more thought. Trh neurons directly innervate *Drosophila* legs (Howard et al., 2019, Curr Biol.) and impact locomotion. Making a claim about social repulsion is a bit too ambitious without further segregating impact on other behaviors like walking that could impact the phenotype observed by the authors.

8) Given that the utility of AR-HMM rests on IR76b experiments which seem to be based on a false premise, it would be best if authors repeat this with one of the myriad of receptors shown to be involved in courtship. Or rather just do AR-HMM on one of the previous datasets.

9) DTW is a cool method but using increased copulation latency of blind flies to show its usability is a bit of a short-sell. It does not in any way increase the credibility of this method and I would rather suggest using DTW to look at longer scale dynamics in previous experiments (e.g. Figure 3 data), provided these are done in the correct genetic background).

Recommendation for improvement: The point of this paper is the method and not the biological findings. But the method loses credibility because there are serious problems with the way the experiments are performed and data is presented. I would recommend the authors chose one strong and clean dataset and take it all the way through their analysis pipeline (selfee->anomaly score->AR-HMM->DTW) instead of using different experiments for explaining different points. Even if the results are less interesting, if the rationale is clear and all data is shown properly this will encourage other people to try out this tool.

*Reviewer #3 (Recommendations for the authors):*

The figures are in a good shape, except where specifically mentioned (and assuming that the final published figures will have better resolution).

Major concerns (2):

1) Reference [9] is critically important for the understanding of how the CLD is implemented in this architecture, however, I find it difficult to access the reference. The authors should provide for an alternative. Adding more visualization of how the CLD is implemented, in Figure 2, would be helpful. The authors should compare the classification performance of Selfee with and without the CLD (for fly courtship behavior in Figure 2). This way the reader could understand the weight of the CLD in the overall architecture. Next, the authors should discuss alternative architectures with CLD and show why the SimSiam has been chosen.

2) How much of the performance of the classification (and possibly other utilities) is lost to strict end-to-end processing? The authors should show the effects of the following pre-processing steps on the t-SNE maps, the confusion matrices, and the F1 scores: a) use the raw frames instead of live-frames; b) perform simple background subtraction; c) create narrow areas of interest around the animals in each frame. Authors can add other steps such as segmentation. The prediction would be that the classification performance will improve greatly, the t-SNE maps should have fewer yet more distinct clusters, and rare behaviors should be classified better than, for example, in Figure 2—figure supplement 5 F.

---

## [Author Response]

Essential revisions:1) Chose one or two strong and clean datasets and take them all the way through the analysis pipeline (selfee->anomaly score->AR-HMM->DTW) instead of using different experiments for explaining different points. We suggest removing IR76b data from the story unless the experiment can be somehow justified.

First, we removed experiments done on Ir76b mutant flies and replaced them with analyses of open-field tests (OFT) performed on mice after chronic immobilization stress (CIS) treatment. We also used this experiment for the whole pipeline (Selfee->anomaly score->ARHMM->DTW), and the results are shown in the last section of Results.

2) Also show Selfee's results on a simpler data set (with *Drosophila* or some other animal) with single-animal video.

We have used Selfee to analyze the video from single-mouse open-field tests. The results are now included in the manuscript.

3) Explain the rationale behind the choice of parameters for "Selfee". Please explain why Selfee performs so poorly without the CLD component. Would a simpler architecture, with the addition of CLD, achieve similar performance as the Selfee?

For neural network design, we followed the ResNet50 architecture and SimSiam framework as cited in the manuscript. For classifications, we used the previously published weighted k-NN classifier and LightGBM classifier without changing parameters. The rationale for choosing other parameters was explained in the manuscript as well as in the following point-by-point response to reviewers’ comments.

The contribution of the CLD loss was discussed in detail in the last paragraph of Siamese convolutional neural networks capture discriminative representations of animal posture, the second section of Results. Further optimization of the neural network architectures was discussed in the Discussion section.

A simpler architecture, with the addition of CLD, may not be able to achieve a similar performance as Selfee. In short, the CLD loss needs a Siamese structure, and SimSiam is the simplest Siamese neural network as far as we know, which has only one ResNet50, one threelayer projector and one term of loss. We don’t envision any Siamese CNNs could be significantly simpler than SimSiam.

4) Explore the behaviors found in different parts of the t-SNE maps and evaluate the effect of the irrelevant features on their distributions. The t-SNE maps display several clusters representing the same behaviors (Figure 2D) which suggests strong overfitting (weak generalization). The authors should at least show some examples taken from different clusters. This would illustrate why Selfee is separating behaviors that humans perceive as a single category.5) Provide a formal comparison with one of the pre-existing methods that would in theory work on the same videos (e.g. JAABA) (Posted 6th Apr 2022).

Thanks for the suggestion. We compared Selfee extracted features with FlyTracker or JAABA engineered features, and we also visualized pose estimation results of animals during intensive social interactions from three widely used toolkits, DeepLabCut, SLEAP and FlyTracker. We found our method could extract features as discriminative as FlyTracker or JAABA features. The outcomes for the pre-existing methods and Selfee were outlined in Table 1.

6) Provide a supplementary table showing full genotype of all experimental animals.

We have provided a detailed resources table before the Methods section.

Reviewer #1 (Recommendations for the authors):Overall I like the paper! I will try to give as many specific, detailed comments as possible in this section.First, to clarify some of the items I listed under "weaknesses" in the public review:(1) Since the paper feels mainly like a Selfee methods paper, I think spending more time describing what exactly Selfee is and how it works would be appropriate. I would recommend putting an expanded version of the Methods section paragraph (lines 925-933) in the main body of the paper, briefly explaining the techniques and existing software as you go through each step. It would give a reader with little machine learning experience a better understanding of the process.

We expanded the method between lines 925 to 933 in the second paragraph of Siamese convolutional neural networks capture discriminative representations of animal posture, the second section of Results. We also explained more about how our pipeline works in the last section of Results and Figure 6.

(2) Efficiency was talked about in the introduction and occasionally brought up when highlighting that Selfee doesn't need large computational/processing power to operate, but I didn't come away with a sense of how much Selfee improves on other approaches in this regard. It would be helpful to be more specific.

We didn’t compare our methods with other self-supervised learning methods formally, because we used our own datasets but not the widely used ImageNet, and training a new model could be computationally expensive. However, as we claimed in the third paragraph of the Discussion, Selfee made self-supervised learning accessible to biology labs by training with only a workstation equipped with one GPU but not multi-GPU servers. We also showed that our method could be transferred from one behavioral assay to another without training.

(3) The comparisons with DeepLabCut make sense -- if it would be straightforward to compare to other software, that might help highlight unique aspects of Selfee a bit more effectively.

We compared Selfee feature extraction with features from FlyTracker or JAABA, two widely used software. We also visualized tracking results of SLEAP and FlyTracker in complement to the DeepLabCut experiment.

(4) I don't usually recommend new experiments when I review papers, but perhaps showing Selfee's results on a simpler data set (with *Drosophila* or some other animal), say with single-animal video, before showing the courtship-type results, could ease a reader into the process more effectively.

We used Selfee with single-mouse OFT of mice after CIS treatment and demonstrated increased thigmotaxis behavior in CIS treated mice. Our results were consistent with previous literature and animal tracking analysis.

(5) It would also be great, if possible, to include a small section that could operate like a guide for a novice user to try Selfee on their own data.

We wrote in the last section of Results to showcase how Selfee works on behavior videos. For further clarity, a more detailed step-by-step guide was uploaded to the GitHub repository.

A few general comments:(1) I am not a machine learning expert myself, so I hope another reviewer could evaluate Selfee's uniqueness compared to other methods, as I am taking that at face value in my review. In particular, (line 17-19), is this true that unsupervised analysis has not been used this way before?

As far as we know, most unsupervised methods required specified equipment or processing procedures, and could not be used in an end-to-end way. For example, MotionMapper requires image registration; ABRS requires “a representative training data set” for dimension reduction, and MoSeq requires depth images but not RGB images. However, we also rewrote lines 17-19 to avoid potential overclaim.

(2) A fair amount of grammar/word usage errors, not really a problem and I assume copy editors will address? (e.g. Fast  Quickly or Rapidly on line 15).

Corrected.

(3) I generally like the figures, but found Figure 2 and Figure 3 pretty difficult to read at that size. Rearranging the panels and/or using larger fonts would help readability considerably.

For Figure 2, we rearranged it, used a larger front, and split it into two separate figures. We also added some annotation to the original Figure 3 (the new Figure 4).

Reviewer #2 (Recommendations for the authors):This work is very relevant but has some serious unresolved issues before warranting further consideration. The overall format of the paper will also have to be massively restructured to make it palatable for a wide, life-science research audience.Major points of concern that need to be addressed:1) In the way it is presented, the rationale behind choice of several of the parameters for "Selfee" are unclear to me. E.g.a. Why do the live frames consist of 3 tandem frames? Is 3 somehow a special number that is easy to compute or represent as an RGB image? Why not 4 with CMYK or a higher number with a distinct colormap as is commonly done for representing depth coding.

There are mainly three reasons. First, as we claimed in the manuscript, we followed the original ResNet50 network architecture and we adopt its public pre-trained weights, so the input channel was set to three. Second, best to our knowledge, we don’t know any examples that use CYMK coding for image inputs of a CNN. As for depth, RGBD is a common choice, but our data did not contain depth information. Last, to implement Turbo transformation whose importance was discussed in the Results, the input has to be an RGB input, which is obvious in Figure 2 and Figure 2—figure supplement 1.

b. The choice of AR-HMM and DTW as post-processing steps after the Selfee analysis is also a bit unclear to me. Especially because the authors do not discuss any alternatives and moreover, they don't even show great promise in the examples later in the manuscript.

We added some discussion about the alternatives, such as UMAP non-linear transformations, HDBSCAN clustering and dynamic time alignment kernel (DTAK).

2) The authors have surprisingly failed to acknowledge and cite the state of the field that directly translates to their novelty claims (Lines 56-59: "There are no pose estimation paradigms that are designed to handle multi-animal contexts"). Both multi-animal DLC (Lauer et al., 2021) and SLEAP (Perreira et al., 2020) were explicitly designed for this purpose. Not to mention a plethora tools for multi-animal behavioral classification, especially during social interactions that provide pretty much all the extracted information obtained using Selfee in this manuscript. Listing a few *Drosophila* specific tools:a. JAABA (Kaabra et al., Nat Methods, 2013)b. Matebook (Ribeiro et al., Cell 2018)c. Flytracker (Hoopfer et al., eLife 2015)

Our original claim was “there is no demonstration of these pose-estimation methods applied to multiple animals of the same color with intensive interactions”, where we intended to note that the current pose-estimation method was not optimized for multiple animals of the same color with intensive interactions. In the original manuscript, we showed the pose estimation results of DLC for our videos on two mice with the same color. In this revision, we added results from SLEAP and FlyTracker. We think that current tracking or pose-estimation algorithms are not suited for intensive social interactions of multiple animals of the same color.

Second, we would like to discuss more about JAABA, the most famous fly tool, aside from what we discussed in the manuscript. JAABA can be generally split into four parts: animal tracking and pose estimation, spatial feature engineering, temporal feature engineering, and classifier. For JAABA, the first part can be replaced FlyTracker, and SLEAP or DeepLabCut with certain modifications. Selfee could be in the same position of the first and the second part. Our method is just an engineering-free feature extractor without specified temporal feature engineering and classifier. The k-NN classifier used in the manuscript was a demonstration that the feature is discriminative. It is weaker compared with the gentle boost machine used in JAABA, or SVM used in FlyTracker, but it has fewer parameters and was a common choice in the previous publications of self-supervised learning. In this version, we showed that our feature was as discriminative as the engineered features from JAABA in some cases.

3) The authors must provide a sup table showing full genotype of all experimental animals. Wherever possible test and controls should be in the same genetic background (this is especially true for loss of function experiments where vision and locomotion is an integral part of the behavior, like the ones done in this manuscript). Legends for videos should also be provided.

A table containing all genetic information was added. Details of genetic background are discussed in previous sections. Legends for videos were provided.

4) Figure 2 demonstrates how good Selfee is at extracting information. To do this, authors have to either look at how data clusters with respect to human annotated labels on a t-sne map or use a k-nn classification step to classify the data. Honestly, the t-sne maps in Figure 2D don't look that promising. K-nn classification looks better with around 70% F1 score. But at this point there should be a formal comparison with one of the pre-exisiting methods that would in theory work on the same videos. I suggest JAABA (or Flytracker+JAABA) since similar labels have been previously extracted using this toolkit. Same holds true for mouse data (2G-I) and especially more important because the F1 scores here are a bit low. Also panels 2F and 2I likely represent cherry picked cases where results are much better than what the scores indicate and hence are not really informative.

We added a comparison between our method and JAABA+FlyTracker on a previously constructed fly dataset, which contained not only features from these two methods but also finegrained scores from two human experts. Our results were comparable with JAABA and FlyTracker. We did not do the same thing on mice videos, because as we showed in Figure 1— figure supplement 3, the tracking result on copulating mice is not informative.

The main reason why the raster plot looked more promising than the F_1_ score is the imbalance of animal behaviors. When we reported F_1_ scores in the original Figure 2 (the new Figure 3), we used the macro average of each behavior, because all behaviors were biologically significant. For classes like copulation attempts or mounting that contained much fewer samples, they are extremely difficult to machine learning models but they contributed to the score equally. However, they did not contribute equally to the appearance of the raster plot due to smaller sample numbers. If we change macro average to micro average (each sample was treated equally), the F1 score of mice behavior would be 86.0% for eight-fold cross-validation. It might make the original Figure 2I (the new Figure 3F) more understandable.

For the original Figure 2F (the new Figure 3C), F_1_ score is 76.14% and mAP is 76.09%, where F_1_ score is a bit higher than seven-fold cross-validation (76.1% vs 72.4%) and mAP is nearly identical (76.1 vs 75.8). However, when conducting this experiment, we used all data for crossvalidation, but not a leave-one-out setup, so a performance boost is not surprised. For Figure the original 2I (the new Figure 3F), F1 sore is 68.1% and mAP is not available for this ensembled classifier. Compared with eight-fold cross-validation, it is only a bit higher (68.1% vs 67.4%), but it is also not supervising because we used an ensemble classifier from crossvalidation experiments.

5) The rationale for definition of "anomaly score" (Figure 3, lines 230-249) is again poorly defended in the text. Could authors clearly define what "meta-representations of frames" means. Also, why restrict k=1 for IES? How is the +-50 frame window chosen for IAS score. Are these empirically derived values after some trial and error, or based on some priors. Since the anomaly score is quite central to the overall utility of this tool, it needs to be proposed in a much more credible way.

Meta-representation means our feature contains spatial features and short temporal features of animal behaviors, and uniforms natural-languages-based human descriptions and engineered features from tracking results in different units (e.g., rad for angle and mm for distance) and scales in a single discriminative Meta-representation (the new Figure 3, Figure 3—figure supplement 1, Table 3).

For IES, which implements a k-NN anomaly detection algorithm, setting k=1 is a trivial and intuitive case of the k-NN algorithm to avoid parameter search of k. We didn’t have any empirical experience for the k value, and we had no validation set for k searching, so k=1 was only a showcase to demonstrate this anomaly detection could work for Selfee features.

For IAS score, as we showed in the original Figure 3—figure supplement 1B (the new Figure 4—figure supplement 1B), the similarity between the central frame and frames that were 50 frames away was quite low, dropping to half of its maximum value. We didn’t test other values, but 50 was a safe choice according to this figure.

6) The unusual subset of neurotransmitter mutants/silenced lines used in Figure 3 data makes one wonder if the provided data is a cherry-picked subset of the full data to make the story easier. I would really appreciate if all data were shown clearly and rationally. The authors really need to establish credibility for persuading the reader to use their method.

As we explained previously, we only used pre-existing videos in our lab which are mainly focused on nutrient sensing and behavior homeostasis. Also, most of the important neurotransmitter mutants are not viable.

7) Trh mutant phenotype is interesting but interpretations of secondary assays warrant some more thought. Trh neurons directly innervate *Drosophila* legs (Howard et al., 2019, Curr Biol.) and impact locomotion. Making a claim about social repulsion is a bit too ambitious without further segregating impact on other behaviors like walking that could impact the phenotype observed by the authors.

We changed our claim from reduced social repulsion to reduced social distance. We don’t think that changed locomotion likely contributed to the reduced social distance. According to Howard et al., Trh mutant flies had faster walking speeds and increased locomotion. In our aggregation behavior assay, increased locomotion could have induced a hyperactive phenotype leading to movements from the top of the triangle chamber to a distant zone of a more comfortable social distance, which we did not observe.

8) Given that the utility of AR-HMM rests on IR76b experiments which seem to be based on a false premise, it would be best if authors repeat this with one of the myriad of receptors shown to be involved in courtship. Or rather just do AR-HMM on one of the previous datasets.

The results of Ir76b were removed. AR-HMM was done with a dataset from mouse open field test.

9) DTW is a cool method but using increased copulation latency of blind flies to show its usability is a bit of a short-sell. It does not in any way increase the credibility of this method and I would rather suggest using DTW to look at longer scale dynamics in previous experiments (e.g. Figure 3 data), provided these are done in the correct genetic background).

We agreed that DTW applied to NorpA mutants was a trivial case, but we intended to demonstrate our method, rather than to make a new biological discovery. In the videos used in the original Figure 3 (the new Figure 4), we discussed male-male interactions of flies. In such an assay we didn’t observe the long-term dynamic of behaviors.

Recommendation for improvement: The point of this paper is the method and not the biological findings. But the method loses credibility because there are serious problems with the way the experiments are performed and data is presented. I would recommend the authors chose one strong and clean dataset and take it all the way through their analysis pipeline (selfee->anomaly score->AR-HMM->DTW) instead of using different experiments for explaining different points. Even if the results are less interesting, if the rationale is clear and all data is shown properly this will encourage other people to try out this tool.

We used different experiments for demonstrations of different methods because it was extremely hard to find an animal that had all short-term anomalies, which can be detected by anomaly detection, micro-dynamic changes, which is the field of AR-HMM, and long-term behavioral difference, suitable for DTW analysis. Most mutant animals we tested only had behavioral changes at one time scale, and strong mutants that were far beyond normal could be easily spotted by human observers and may be too trivial to demonstrate the power of automated self-supervised behavior analysis. We went through our Selfee pipeline with mice OFT assays.

We didn’t find any short-term anomaly in CIS-treated mice, but we found their increased thigmotaxis behavior could be proved by animal tracking.

Reviewer #3 (Recommendations for the authors):The figures are in a good shape, except where specifically mentioned (and assuming that the final published figures will have better resolution).Major concerns (2):1) Reference [9] is critically important for the understanding of how the CLD is implemented in this architecture, however, I find it difficult to access the reference. The authors should provide for an alternative. Adding more visualization of how the CLD is implemented, in Figure 2, would be helpful. The authors should compare the classification performance of Selfee with and without the CLD (for fly courtship behavior in Figure 2). This way the reader could understand the weight of the CLD in the overall architecture. Next, the authors should discuss alternative architectures with CLD and show why the SimSiam has been chosen.

The CLD paper can be accessed from two sources:

UCB official website http://people.eecs.berkeley.edu/~xdwang/papers/CVPR2021_CLD.pdf and the preprint version https://arxiv.org/abs/2008.03813v4 .

The GitHub repository is at:

https://github.com/frank-xwang/CLD-UnsupervisedLearning.

The comparison results between with and without CLD were in the original Figure 2—figure supplement 6 (the new Table 2).

In our paper, we simply chose a simpler model SimSiam but not a more complex one like BYOL. In our GitHub repository, we implemented the BYOL architecture and it would work. However, we were afraid that it may cause confusion to biological researchers if more neural network structure was talked about. We intended to focus on self-supervised learning but not one specific network design.

2) How much of the performance of the classification (and possibly other utilities) is lost to strict end-to-end processing? The authors should show the effects of the following pre-processing steps on the t-SNE maps, the confusion matrices, and the F1 scores: a) use the raw frames instead of live-frames; b) perform simple background subtraction; c) create narrow areas of interest around the animals in each frame. Authors can add other steps such as segmentation. The prediction would be that the classification performance will improve greatly, the t-SNE maps should have fewer yet more distinct clusters, and rare behaviors should be classified better than, for example, in Figure 2—figure supplement 5 F.

For raw frames instead of live-frames, results were added in the new Table 3, and there was around one to three percent performance loss. For background subtraction, we actually used it for all mice videos and fly videos when comparisons with JAABA and FlyTracker were made.

We found that our method would not work without background subtraction for mice mating videos because the beddings caused a lot of distraction. Segmentation of two animals was actually very difficult. Because the relative position between two animals was difficult to preserve after the segmentation, and segmenting intensively interacting animals was also very hard. Even more, thigmotaxis behavior became hard to identify after segmentation, which was very important to evaluate mice anxiety. Therefore, we did not implement animal segmentation.